# How does analyst coverage influence corporate social responsibility (CSR)? The governance- and information-based perspectives

Yan Liu[1], Xiuhong Du[2]*

**1** Sinopec Marketing Shandong Company, Jinan, Shandong, China, **2** School of Economics, Nanjing Audit University, Nanjing, Jiangsu, China

* duxiuhong@nau.edu.cn

**Data Availability Statement:** All relevant data are within the manuscript and its Supporting information files.

**Funding:** The study was funded by the Belt and Road National Audit Research Center of Nanjing

## Abstract

Based on a sample of Chinese public manufacturing firms, this study empirically investigates whether and how analyst coverage drives corporate social responsibility (CSR) under different governance or information conditions. The results show that firms with greater analyst coverage take more social responsibility, representing magnified concerns and better CSR visibility for legitimacy and reputation. This relationship could be strengthened under high governance condition (high institutional ownership ratio, none CEO duality, low executive ownership) or low information situation (high earnings management and low accounting conservatism). These findings provide new evidence of information-based mechanism underlying the promotions of CSR in imperfect information environments.

## Introduction

Nowadays, corporate social responsibility (CSR) has been the common focus of government, society, and the firms [1–4]. CSR not only belongs to the ethic part of the firms, but also delivers financial implications [5]. When making decisions, stakeholders increasingly weigh CSR in addition to traditional financial information [6]. Across the world, many firms have strategically responded to these shifting expectations by enthusiastically embracing CSR practices. Given the great importance of CSR for firms and society, scholars have devoted to investigate the diverse individual, organizational, and formal and informal institutional determinants of CSR. For instance, CEO political ideologies and incentives [7], corporate governance [8], culture [9], institutional pressure [10], industrial environments [11], and religion [12] could significantly influence CSR performance. Recently, a strand of attention has been paid to disclose the impact of external analysts' attention on a firm's CSR and its underlying mechanisms [13].

Financial analysts are certified experts at providing evaluations and recommendations about their followed firms [14]. They serve as key information intermediaries to improve organizational visibility in the capital market by gathering, processing, and disseminating relevant market and firm-specific information [15]. While there is some evidence that pressures from

Audit University (No. A2010560013 to XH Du). The funders had no role in study design, data collection and analysis, decision to publish, or preparation of the manuscript.

**Competing interests:** The authors have declared that no competing interests exist.

analysts' push managers to beat the short-period earnings targets at the expense of long-period economic value, analysts are increasingly favoring and paying attention to CSR-related information [6, 16]. Considering CSR as a value-enhancing activity, analysts are more willing to recommend to invest in firms with better CSR.

Previous paper focuses on the monitor function of analyst coverage on CSR [15, 17], while giving little attention to the information function of analyst coverage, especially making comparisons between the monitor and information functions. For example, Zhang et al. (2015) find analysts can improve corporate philanthropy through promoting CSR, and the results are more pronounced for nonstate-owned enterprises (non-SOEs) [15]. Hinze and Sump (2019) systematizes the research on the relationship between analyst coverage and CSR [5]. Hu et al. (2021) emphasizes the governance mechanism of financial analysts, and examines the moderating function of corporate governance [17]. Naqvi et al. (2021) suggests that CSR decreases information asymmetry, and analyst coverage moderates the negative impact [18]. These studies offer evidence that financial analysts could exert influence on CSR through their governance or information related functions. In practical, investing in CSR activities is dependent on motivations of insiders such as the CEO and the board. Their evaluations on the information from analysts towards CSR are often affected by their power or discretionary decision-making. The concentration of the power makes CEO has the negotiating advantage among the parties and be less likely to cater to the expectations of stakeholders. Compared to the retail investors, institutional investors not only pay greater attention to various stakeholders (e.g. analysts), but also express serious concerns about societal expectations (e.g. CSR) in the oversight of managerial decision-making processes. Like CEO duality, concentrated executive ownership enlarges the power and influence of manager and could lead to the similar outcome of dual CEOs. Thus, CEO duality, institutional ownership, or executive ownership might be the driving forces of analysts on corporate CSR to a certain extent. Additionally, the information heterogeneity, such as various accounting information characters (e.g. real earnings management and accounting conservatism), may result in a different impact of analyst coverage on CSR, which is scarcely empirically examined in the prior studies. Meanwhile, both real earnings management and accounting conservatism reflect the reliable degree of external financial information [19, 20]. Real earnings management emphasizes whether the accounting information is manipulated through real activities, such as adjusting production and producing cost. Accounting conservatism positively relates to the timeliness of confirming bad news, while negatively collated to good news. The process of considering financial analysts' opinions into CSR behaviors cannot ignore the necessary conditions which comprise reactions from the capital market. It is necessary to specially disclose the role of external financial information in corporate strategies towards CSR investments. While the characteristics of external financial information is significant different from corporate governance system, the comparisons of the effects of corporate governance and external financial information are also relevant.

This study seeks to fill the literature gap by developing knowledge about the influence of analysts on a firm's CSR performance under different governance and information conditions. First, how does the structure of corporate governance influence the relationship between analyst coverage and CSR? Second, how do the characteristics of firm financial information influence the relationship between analyst coverage and CSR? Third, is there any differences between the moderate effect of governance and information condition on the relationship between financial coverage and CSR. The intriguing and solid research findings confirm the theoretical hypotheses and suggest that analyst coverage has positive effects on CSR performance via information mechanism. Interestingly, firms covered by more analysts are more aggressive in pursuing socially responsible activities when institutional ownership is high, or when executive ownership is low. The absence of CEO duality also plays a positive moderating

role. Additionally, we examine the moderating role of information characteristics, including real earnings management and accounting conservatism condition. Real earnings management enlarges the positive effect of financial analysts on CSR. The less of accounting conservatism also plays an enhancing moderating effect and promotes analyst coverage to make more positive impact on firms' socially responsible activities. It is observed that analyst coverage complements with the governance mechanism, while it substitutes with the information mechanism to make a difference on CSR. In other words, the information moderators do not work on the same direction with the governance moderators. These results support our hypotheses that with regards to CSR, the information effect is stronger than the pressure effect.

## Theoretical background and hypotheses

**Analyst coverage and CSR.** Financial analysts are professional experts who collect information, evaluate, and make forecasts about firms they follow, and provide investment recommendations to current and potential investors [21]. The extant literature has implied two distinct effects that analysts can make towards firms' strategic decisions. One strand, named "pressure effect", stresses the negative aspect of analyst coverage [22]. Financial analysts pay too much attention to the short-term performance of the firms, which results in managers cutting investment in long-term programs. Managers, especially of public or international firms, become more myopic when they receive higher attention from external forces including financial analysts [23]. Another strand, named the "information effect", emphasizes the bright side of analyst coverage [12]. By relieving information asymmetry, analyst coverage could increase organizational visibility and inhibit managers from making myopic decisions and wasteful discretionary spending [6, 15]. By this logic, we argue that managers may have stronger motivations to achieve better CSR when their firms are tracked by more financial analysts.

First, analyst coverage could promote CSR by relieving agency problem and averting managers' opportunistic behaviors. For routine tasks, analysts constantly monitor managers by analyzing, processing, and transmitting message about firm performance and probing into business strategies [24]. Firms with higher analyst coverage often attract greater attention and scrutiny from investors. Corporate strategies as to whether and how to make socially responsible investments are not only affected by the expected returns but also by other careful considerations. Despite uncertainty around CSR, external concerns from analyst could superintend the managers' behavior and propel them to spend time, effort, and allocate resources to fulfill their duty to construct a better social community beyond the official legal obligations. The reduction in CSR investment, once observed by analysts, could result in considerable negative outcome of firm market value. To minimize the possibility of getting adverse feedbacks, managers would be less aggressive in reducing CSR investment and make earnest endeavors to achieve better CSR performance toward which analysts have favorable attitudes.

Second, the attention from analysts could ease the information asymmetry problem between firms and related parties. CSR is widely regarded as an important corporate behavior for organizational legitimacy and long-term success. But it is complex for general investors to estimate its real and intrinsic value due to the multidimensional nature of CSR. The valuation for firms will be discounted with incompletely conveyed CSR information [25]. Analysts could recognize the potential of CSR and factor it into investment recommendations [6]. Stock recommendations from financial analysts which represent a supposedly unbiased third party are valued by investors, especially in the developing country-level external governance environment [26]. While analysts not only pay attention to accounting fundamentals but also to other relevant professional and private information in poor information environments, analyst coverage is more influential in boosting CSR by exhibiting accurate assessment of social

investment [15]. CSR is the key component of organizational sustainability, and CSR-related policies have become critical non-financial information for analysts when they prepare earnings forecasts. More analyst coverage could reduce information asymmetry and help to reasonably value corporate CSR. Furthermore, the growing public attention to sustainability calls for corporate timely responses to prosocial demands and expectations from stakeholders. Analysts often show optimistic views when a firm has better social performance. Firms with higher analyst coverage consequently are often positively evaluated by investors when they perform better CSR. Considering the significant impact of analysts on the valuation of investors [26], managers will be pushed to take account of CSR when the related firms are followed and covered by analysts.

Therefore, we predict that:

H1: Analyst coverage positively relates to firms' CSR performance.

**The role of governance characteristics.** *Moderating effect of CEO duality.* The effectiveness of analyst monitoring on CSR is also associated with the appearance of CEO duality. (1) CEO duality increases the information asymmetries among stakeholders [27]. CEO duality places barriers in the way of analyst coverage. This impaired accountability jeopardizes analysts' judgements and restricts the ameliorative impact of their monitor function on information asymmetry. (2) CEO duality roles provide more room for opportunistic behaviors. Dual CEOs have more incentives to express serious concern about private interest or short-term goals and reduce the resources allocating to CSR activities [28]. (3) Duality endows CEOs with enormous power (Rashid, 2013), decreasing the influence of outside analysts on inside CSR. When a CEO also takes the position of Chairman, the power of CEO is enhanced by concentration. Thus, the dual CEOs are less worried about the monitor function and discipline effect of financial analysts. The concentrated power increases the CEO's negotiating advantage among stakeholders [29], and makes them less likely to pay attention to analysts' opinions, such as prioritizing CSR. A dual leadership structure reduces checks and scrutiny of these CEOs, and enhances the unbalance between CEOs and their boards, making CEOs less accountable to all stakeholders. Hence, they are less worried about pressure from outsiders, including financial analysts. In other words, the function of financial analysts on CSR needs the cooperation of the CEO, while CEO duality decreases the CEOs' incentives of behaving in the preference ways of the analysts.

As a result, dual CEOs restrain good corporate governance practices and often neutralize the positive effects of analyst coverage on CSR. Thus, we predict that:

H2: CEO duality mitigates the impact of analyst coverage on CSR.

*Moderating effect of institutional investor.* Risk is a significant factor considered by the institutional investors. Institutional investors are reluctant to pick and invest in the risky stocks, ceteris paribus [30]. Usually, firms with strong CSR have less systematic risk and greater growth opportunity [31, 32]. As a result, institutional investors favor corporate engagements in CSR.

As important shareholders, institutional owners are more concerned about and actively involved in organizational strategic decisions than non-institutional stockholders. The rapid growth of stakeholder demands, fund flow benefits, and risk reduction are the reasons that institutional investors prioritize firms with social benefit and sustainable goals. Being a strong symbol of prosocial responsibility, CSR naturally is regarded as an important indicator for investment decisions. Institutional investors often have a shared understanding of the value of CSR with analysts for the following firms. Accordingly, institutional investors are inclined to

seek out higher-CSR-performing firms and enlarge the related share of their portfolios when analysts are prioritizing CSR [33]. Therefore, the firms with high institutional ownership have more motivations to engage in CSR when analysts are encouraging corporate investments in CSR. Meanwhile, compared to the scatter investors, the institutional investors have the ability to hold the stock for a longer time, and hence are probably more care about the long-term activities [34, 35]. They have greater tolerance for decreasing of short-term profits [35]. The more tolerance investors have, the less pressure analyst average conducts, and the more contemplations and motivations managers will be inspired to take the opportunities and bear the uncertainty of CSR to go along with the opinions from analysts.

Accordingly, we predict that:

H3: Institutional ownership positively moderates the relationship between analyst coverage and CSR.

*Moderating effect of executive ownership*. Concentrated executive ownership enlarges the power and influence of managers, while weakens the monitoring function of financial analysts over them [28]. Because the lack of monitor efficiency, executive ownership may exert significant negative influence on the effectiveness of analyst coverage on CSR.

First, with high executive ownership, a firm's accountability is less of guarantee due to the weakened monitor effect from outsiders. The analysts' monitoring is no longer valid to the internal decisions about the strategies and policies related to social activities. Second, high executive ownership also generates agency problem [36]. Greater executive ownership facilitates the entrenchment of top managers and shields them from the punishment of self-benefit behaviors. With high executive ownership, entrenched managers care less about external investors and are more reluctant to devote resource in social construction projects [25]. Third, the presence of high executive ownership makes strategies more dependent on the preference of managers. The value guidance from analysts is likely to be given a smaller weight. Firms with high executive ownership often have less motivations to engage in CSR although analysts are encouraging corporate investments in CSR. For example, firms cut off charitable donation after an increase in executive ownership [37]. Thus, a high level of executive ownership will lead to the constraint of analysts and pay less attention to analysts' opinions, such as prioritizing CSR, and we predict that:

H4: Executive ownership negatively moderates the relationship between analyst coverage and CSR.

**The role of information characteristics.** *Moderating effect of real earnings management*. Earnings management could adjust accounting numbers without interrupting real activities [19, 20]. With improving accounting standards, the legal range of earnings management has been reduced. More and more managers alter earnings via real operating projects and activities. Even it is not explicitly mentioned, the diversity of the performance in CSR may also be a subtle form of real earnings management. Managers can choose to devote resource to CSR according to the real earnings performance. When real earnings outstrip expected earnings, the managers may choose to invest more in CSR. When earnings fall under the expected value, managers may cut their CSR budget.

On the one hand, the severe potential harm of real earnings management makes financial analysts more alert. However, firms with real earnings management are less likely convinced by professional accounting information institutions [38], such as financial analysts. With the coverage of financial analysts, the adjustment of CSR is more detectable. We infer that firms, with a higher degree of real earnings management, are less likely to reduce the investment in

CSR, because they are more suspicious by financial analyst. The firms are inclined to behave in a way of devoting in CSR to blur the potential unfavorable impression from financial analyst, caused by real earnings management.

On the other hand, real earnings management enlarges the function space of financial analysts on the firms. Real earnings management increases information opacity, accounting information prediction ability and capital cost, and financial constraints [38]. Therefore, the information function of the analysts on the firms are more obvious in the firms with real earnings management. In other words, financial analysts play a more profound role on CSR in firms with larger degree of real earnings management. Firms naturally have more motivations to engage in CSR to go along with analysts when they have special preference to better CSR performance.

H5: Real earnings management positively moderates the relationship between analyst coverage and CSR.

*Moderating effect of accounting conservatism*. The speed of earnings response to bad or good news is a key indicator of accounting conservatism [39]. Specifically, the timeliness of bad news stands for the conservatism attitude, while the timeliness of good news proxies for the opposite. Accounting conservatism mitigates the interest conflicts among the agent, the owners, and the debtors. Accounting conservatism not only reduces the overestimated degree of the net asset value, but also promotes managers to recognize bad news as soon as possible, reducing the value transfer from the debtors to large stockholders. It shapes the behavior of the agency and insiders [40]. In other words, accounting conservatism may be substitute with financial analysts by regulating the agent parties. The value guidance from analysts is likely to be given a smaller weight in this event. Although analysts have special preference to CSR, managers often choose to turn a blind eye to these opinions. Thus, analyst coverage makes less influence on CSR in the firms with more accounting conservatism.

H6.1: The timeliness of bad news (more accounting conservatism) negatively moderates the relationship between analyst coverage and CSR.

H6.2: The timeliness of good news (less accounting conservatism) positively moderates the relationship between analyst coverage and CSR.

The conceptual structure is shown in Fig 1.

## Methodology

**Data and samples.** The sample includes publicly listed manufacturing firms in Chinese Stock "A" markets and starts from 2010 and ends in 2018. Manufacturing firms have been chosen because they are the mainspring of economy development and the main sources of air, land, and water pollution in China. The research on their CSR is quite relevant for the sustainability of the country. Although the CSRC (China Securities Regulatory Commission) began requesting listed firms' disclosure of CSR information in 2008, there was scarce voluntary disclosure of CSR reports even through 2009. Thus, our sample period begins in 2010 when the social responsibility disclosure in China's capital market started to be normalized. Analyst coverage data, CSR scores, and other firm specific variables are obtained from the website of China Stock Market and Accounting Research (CSMAR) database, and the client software of Wind Financial (WIND) database. To decrease the effect of outliers on results, we further winsorize the continuous variables at the 99th percentile [13, 19]. The sample in the baseline regressions contains 10896 firm-year observations.

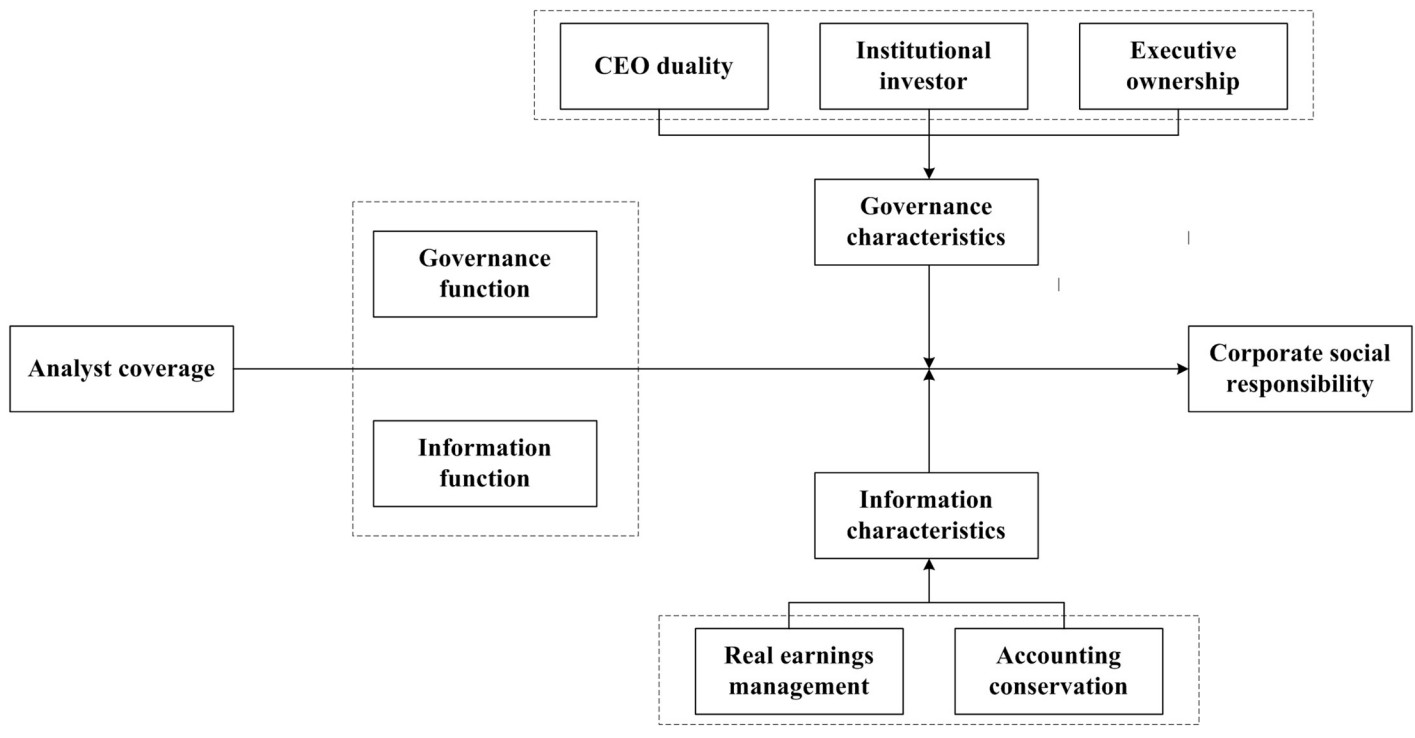

**Fig 1. The conceptual structure.**

**Measurements.** *Dependent variable*. HEXUN CSR Ranking is applied to measure a firm's social responsibility performance, which is the dependent variable. HEXUN CSR Ranking is widely applied in the research of CSR in China [12]. The evaluation index is built according to firms' CSR report and financial report, and provides a multi-dimensional rating consisting of five pillars of responsibility performance towards: (1) shareholders, (2) employees, (3) suppliers, customers and consumer rights, (4) environment, and (5) society. These five dimensions include 13 second-class and 38 third-class indices. The overall CSR score is calculated on account of the corresponding weight for each index. A higher CSR score proxies a better CSR performance. To provide a panoramic analysis, a total evaluation score (*CSR_T*) and five sub-dimension scores in the next year are finally used the following analysis (*CSR_Sha* for shareholders, *CSR_Emp* for employees, *CSR_SCC* for suppliers, customers and consumer rights, *CSR_Env* for environment, and *CSR_Soc* for society respectively). All dependent variables are lagged one year to partly mitigate the potential endogeneity.

*Independent variable*. As the independent variable, analyst coverage is measured by the number of analyst teams that issue analysis reports for a firm in each fiscal year [21, 22, 41]. It relies on the fact that all analyst teams issue at least one recommendation or forecast report for the following firm during the related fiscal year period. To correct for the potential skewedness, natural logarithm (one plus) is used to calculate the finial value (*Coverage*).

*Moderators*. There are 5 moderators in this study: CEO duality, institutional ownership, executive ownership, real earnings management, accounting conservatism. CEO duality (*CEODu*) is a dummy variable, and it equals 1 if the position of chairman and CEO are held by one person in a firm and 0 otherwise. Institutional ownership (*InOwner*) is measured by the ratio of stocks owned by institutional investors [27]. Executive ownership (*ExOwner*) is calculated by the shareholding ratio of top managers [35]. We predict real earnings management

(*RES*), according to Cheng et al. (2016) [42]. Accounting conservatism is calculated by *C_score and G_score*, according to Khan and Watts (2009) [43].

*Control variables*. Following the previous literature, a series of variables are controlled to eliminate individual differences which are likely to influence a firm's corporate social responsibility [12, 25]. Specifically, the control variables include: the size of firm (Size: the natural logarithm value of assets), fixed assets (PPE: net properties, plants, and equipment scaled divided by assets), the ratio of leverage (Lev: the ratio of net leverage to assets), the concentration degree of ownership (Top10h: the shareholding ratio of all the top 10 stockholders), whether paid dividend or not (Div: whether firm paid dividend in the fiscal year, if it did, when record Div as 1, or as 0), firm value (TobinQ: market value scaled by the book value).

*Regression models*. To investigate how analyst coverage influences firms' CSR, we estimate following regression (Eq 1) using panel data analysis. Following the related literature of analyst coverage, we apply 2SLS to mitigate the concern for endogeneity [22, 41]. We care for the coefficient of *Coverage*, and predict it to be positive according to H1.

$$CSR\_X_{i,t} = \alpha_0 + \beta_1 Coverage_{i,t-1} + \beta_2 Size_{i,t-1} + \beta_3 PPE_{i,t-1} + \beta_4 Lev_{i,t-1} + \beta_5 Top10h_{i,t-1} + \beta_6 Div_{i,t-1} + \beta_7 TobinQ_{i,t-1} + Year_t + Firm_i + \varepsilon_{i,t} \tag{1}$$

To investigate the moderating effect of managers' duality, we employ following regression (Eq 2) using panel data analysis. *CEODu* is managers' duality. We care for the coefficient of the interaction of *CEODu *Coverage* and predict it to be negative according to H2.

$$CSR\_X_{i,t} = \alpha_0 + \beta_1 Coverage_{i,t-1} + \beta_2 CEODu_{i,t-1} + \beta_3 CEODu_{i,t-1}*Coverage_{i,t-1} + \beta_4 Size_{i,t-1} + \beta_5 PPE_{i,t-1} + \beta_6 Lev_{i,t-1} + \beta_7 Top10h_{i,t-1} + \beta_8 Div_{i,t-1} + \beta_9 TobinQ_{i,t-1} + Year_t + Firm_i + \varepsilon_{i,t} \tag{2}$$

To investigate the moderating effect of institutional ownership, we employ following regression (Eq 3) using panel data analysis. *InOwner* is institutional ownership. We care for the coefficient of the interaction of *InOwner*Coverage* and predict it to be positive according to H3.

$$CSR\_X_{i,t} = \alpha_0 + \beta_1 Coverage_{i,t-1} + \beta_2 InOwner_{i,t-1} + \beta_3 InOwner_{i,t-1}*Coverage_{i,t-1} + \beta_4 Size_{i,t-1} + \beta_5 PPE_{i,t-1} + \beta_6 Lev_{i,t-1} + \beta_7 Top10h_{i,t-1} + \beta_8 Div_{i,t-1} + \beta_9 TobinQ_{i,t-1} + Year_t + Firm_i + \varepsilon_{i,t} \tag{3}$$

To investigate the moderating effect of executive ownership, we employ following regression (Eq 4) using panel data analysis. *ExOwner* is executive ownership. We care for the coefficient of the interaction of *ExOwner *Coverage* and predict it to be negative according to H4.

$$CSR\_X_{i,t} = \alpha_0 + \beta_1 Coverage_{i,t-1} + \beta_2 ExOwner_{i,t-1} + \beta_3 ExOwner_{i,t-1}*Coverage_{i,t-1} + \beta_4 Size_{i,t-1} + \beta_5 PPE_{i,t-1} + \beta_6 Lev_{i,t-1} + \beta_7 Top10h_{i,t-1} + \beta_8 Div_{i,t-1} + \beta_9 TobinQ_{i,t-1} + Year_t + Firm_i + \varepsilon_{i,t} \tag{4}$$

To investigate the moderating effect of real earnings management, we employ following regression (Eq 5) using panel data analysis. *RES* is real earnings management. We care for the coefficient of the interaction of *RES *Coverage* and predict it to be negative according to H5.

$$CSR\_X_{i,t} = \alpha_0 + \beta_1 Coverage_{i,t-1} + \beta_2 RES_{i,t-1} + \beta_3 RES_{i,t-1}*Coverage_{i,t-1} + \beta_4 Size_{i,t-1} + \beta_5 PPE_{i,t-1} + \beta_6 Lev_{i,t-1} + \beta_7 Top10h_{i,t-1} + \beta_8 Div_{i,t-1} + \beta_9 TobinQ_{i,t-1} + Year_t + Firm_i + \varepsilon_{i,t} \tag{5}$$

In order to predict RES, we employ Eqs (6), (7) and (8).

$$CFO_{i,t}/Size_{i,t-1} = \alpha_0 + \beta_1 1/Size_{i,t-1} + \beta_2 REV_{i,t}/Size_{i,t-1} + \beta_3 \Delta REV_{i,t-1}/Size_{i,t-1} + \varepsilon_{i,t} \quad (6)$$

$$PROD_{i,t}/Size_{i,t-1} = \alpha_0 + \beta_1 1/Size_{i,t-1} + \beta_2 REV_{i,t}/Size_{i,t-1} + \beta_3 \Delta REV_{i,t-1}/Size_{i,t-1} + \varepsilon_{i,t} \quad (7)$$

$$DISEXP_{i,t}/Size_{i,t-1} = \alpha_0 + \beta_1 1/Size_{i,t-1} + \beta_2 \Delta REV_{i,t-1}/Size_{i,t-1} + \varepsilon_{i,t} \quad (8)$$

$$RES_{i,t} = -1*A\_CFO + A\_PROD_{i,t} + A\_DISEXP_{i,t} \quad (9)$$

$$APC_{i,t} = A\_PROD_{i,t} \quad (10)$$

*CFO* stands for the net operational cash flow. *PROD* is the production cost and equals to the change of operating cost and inventory. *DISEXP* proxies for the discretionary expense and equals to the sum of sale and management fees. *REV* is the operating income, while *ΔREV* is the change of operating income. We operate the regression of Eqs (6), (7) and (8) by industry and year, to calculate the residual as the abnormal value. The sum of the abnormal value of $(A\_PROD_{i,t})$, the abnormal value of $(A\_DISEXP)$, and the abnormal value of $(A\_CFO)$ multiplied by -1 is the value of real earnings management (*RES*), as shown in Eq (9). The abnormal value of $(A\_PROD_{i,t})$ proxies *APC*, stands for the degree to which the related firms manage production to adjust earnings, as shown in Eq (10).

To investigate the moderating effect of real earnings management in the form of managing production, we employ following regression (Eq 11) using panel data analysis. *APC* is real earnings management in the form of managing production. We care for the coefficient of the interaction of *RES* *Coverage* and predict it to be negative according to H5.

$$CSR\_X_{i,t} = \alpha_0 + \beta_1 Coverage_{i,t-1} + \beta_2 APC_{i,t-1} + \beta_3 APC_{i,t-1}*Coverage_{i,t-1} + \beta_4 Size_{i,t-1} + \beta_5 PPE_{i,t-1}$$
$$+ \beta_6 Lev_{i,t-1} + \beta_7 Top10h_{i,t-1} + \beta_8 Div_{i,t-1} + \beta_9 TobinQ_{i,t-1} + Year_t + Firm_i + \varepsilon_{i,t} \quad (11)$$

To investigate the moderating effect of accounting conservatism, we employ following regression (Eq 12) using panel data analysis. *C_score* is the timeliness to bad news. The larger the value of *C_score*, the more accounting conservatism it is. We care for the coefficient of the interaction of *C_score*Coverage* and predict it to be negative according to H6.

$$CSR\_X_{i,t} = \alpha_0 + \beta_1 Coverage_{i,t-1} + \beta_2 C\_score_{i,t-1} + \beta_3 C\_score_{i,t-1}*Coverage_{i,t-1} + \beta_4 Size_{i,t-1}$$
$$+ \beta_5 PPE_{i,t-1} + \beta_6 Lev_{i,t-1} + \beta_7 Top10h_{i,t-1} + \beta_8 Div_{i,t-1} + \beta_9 TobinQ_{i,t-1} + Year_t + Firm_i + \varepsilon_{i,t} \quad (12)$$

To investigate the moderating effect of accounting conservatism, we further employ following regression (Eq 13) using panel data analysis. *G_score* is timeliness of good news. The larger the value of *G_score*, the less accounting conservatism it is. We care for the coefficient of the interaction of *G_score*Coverage* and predict it to be positive according to H6.

$$CSR\_X_{i,t} = \alpha_0 + \beta_1 Coverage_{i,t-1} + \beta_2 G\_score_{i,t-1} + \beta_3 G\_score_{i,t-1} * Coverage_{i,t-1} + \beta_4 Size_{i,t-1} + \beta_5 PPE_{i,t-1}$$
$$+ \beta_6 Lev_{i,t-1} + \beta_7 Top10h_{i,t-1} + \beta_8 Div_{i,t-1} + \beta_9 TobinQ_{i,t-1} + Year_t + Firm_i + e_{i,t} \quad (13)$$

We apply the following models to predict *C_score and G_score*. *EPS* is the earnings per share. *P* is the price of the stock at the very beginning of the year. *R* is the return of the stock. *D* is a dummy variable. *D* equals to 1 if *R* is negative, and to 0 vise versa. *MB* is the ratio of market

value to asset.

$$EPS_{i,t}/P_{i,t} = \alpha_0 + \beta_1 D_{i,t} + \beta_2 R_{i,t} + \beta_3 D_{i,t}*R_{i,t} + \varepsilon_{i,t} \qquad (14)$$

$$C\_score_{i,t} = \beta_3 = \mu_1 + \mu_2 Size_{i,t} + \mu_3 Lev_{i,t} + \mu_4 MB_{i,t} \qquad (15)$$

$$G\_score_{i,t} = \beta_2 = \gamma_1 + \gamma_2 Size_{i,t} + \gamma_3 Lev_{i,t} + \gamma_4 MB_{i,t} \qquad (16)$$

$$EPS_{i,t}/P_{i,t} = \alpha_0 + \beta_1 D_{i,t} + (\gamma_1 + \gamma_2 Size_{i,t} + \gamma_3 Lev_{i,t} + \gamma_4 MB_{i,t})R_{i,t} + (\mu_1 + \mu_2 Size_{i,t} + \mu_3 Lev_{i,t}$$
$$+ \mu_4 MB_{i,t})D_{i,t}*R_{i,t} + \varepsilon_{i,t} (17)$$

We do the regression of Eq (17) by year and industry. We put the coefficient of $\gamma_1\text{~}\gamma_4$ and $\mu_1\text{~}\mu_4$ back to Eqs (15) and (16) to calculate $C\_score_{i,t}$ and $G\_score$.

## Results

### Main results

Table 1 shows the descriptive statistics. Table 2 reports the Pearson correlation matrix for the variables in empirical models. None of the correlation coefficients are larger than or equal to 0.70 and within acceptable limits.

To investigate how analyst coverage influences firms' CSR, we employ a serials of regression using panel data analysis. The coefficients between CSR and analyst coverage is 0.341 (0.430, 0.189, 0.181, 0.166, and 0.116 for the five sub-dimensions respectively), which are positive and significant. These preliminary results show that analyst coverage and CSR are positively related, supporting Hypothesis 1.

As shown in Table 3, the corresponding coefficients of *Coverage* are 1.410 (S.E. = 0.212, $p < 0.01$), 0.997 (S.E. = 0.084, $p < 0.01$), 0.110 (S.E. = 0.038, $p < 0.01$), 0.129 (S.E. = 0.060, $p < 0.01$), 0.141 (S.E. = 0.069, $p < 0.01$), and 0.033 (S.E. = 0.051, *n.s.*) respectively. The results show that the more financial analysts a firm is followed, the better CSR it performs. In fact, the society dimension of HEXUN CSR Ranking refers to the contribution of tax and donation amount. Relative to other dimensions, investing in philanthropy or paying more taxes seems be an immediate discount on financial profits. Although philanthropic activities may help firms to get reputational capital and improve long term performance, investments in social dimension of CSR are not preferential when financial constraints exist [15]. On the whole, analyst coverage contributes to the improvements of CSR. The empirical evidences support that information mechanism of financial analyst probably dominates the pressure mechanism, making a positive impact on CSR in the end. Therefore, H1 is supported.

To mitigate the potential concern of multicollinearity, moderators are mean-centered before the creation of interaction terms [14]. As shown in Table 4, the coefficients of *Coverage* (1.396, $p < 0.01$; 0.993, $p < 0.01$; 0.108, $p < 0.01$; 0.127, $p < 0.05$; 0.136, $p < 0.05$; 0.032, *n.s.*) almost remain positive and significant. The coefficients of interaction term (*CEODu\*Coverage*) are all negatively significant (-1.379, S.E. = 0.360, $p < 0.01$; -0.237, S.E. = 0.143, $p < 0.1$; -0.185, S.E. = 0.064, $p < 0.01$; -0.289, S.E. = 0.110, $p < 0.01$; -0.316, S.E. = 0.115, $p < 0.01$; -0.308, S.E. = 0.087, $p < 0.01$). which exhibits that the positive relationships between analyst coverage and CSR are weakened when the chairman and CEO are the same person. Offering the greater power to a person implies greater degree of decision discretion. Thus, the presence of CEO duality often reflects poorer monitoring, resulting the negligence of socially responsible activities [36]. Under the situation of concentrated authority, the role of analyst coverage

**Table 1. Descriptive statistics.**

| Variable | Observations | Mean | S.D. | Minimum | Maximum |
|---|---|---|---|---|---|
| CSR_T | 10,896 | 23.98 | 16.51 | -15.43 | 90.83 |
| CSR_Sha | 10,896 | 14.01 | 6.73 | -12.79 | 28.18 |
| CSR_Emp | 10,896 | 2.55 | 3.11 | -0.17 | 15.00 |
| CSR_SCC | 10,896 | 1.65 | 4.66 | 0.00 | 20.00 |
| CSR_Env | 10,896 | 1.84 | 5.33 | 0.00 | 30.00 |
| CSR_Soc | 10,896 | 3.93 | 3.68 | -15.00 | 23.90 |
| Coverage | 10,896 | 1.64 | 1.11 | 0.00 | 4.19 |
| Size | 10,896 | 21.59 | 0.97 | 19.17 | 24.76 |
| PPE | 10,896 | 0.13 | 0.12 | 0.00 | 0.58 |
| Lev | 10,896 | 0.33 | 0.20 | 0.01 | 1.00 |
| Top10h | 10,896 | 58.64 | 14.56 | 22.13 | 89.16 |
| Div | 10,896 | 0.77 | 0.42 | 0.00 | 1.00 |
| TobinQ | 10,896 | 2.65 | 1.97 | 0.27 | 12.24 |
| CEODu | 10,820 | 0.298 | 0.457 | 0.00 | 1.00 |
| InOwner | 10,724 | 0.35 | 0.23 | 0.00 | 0.85 |
| ExOwner | 10,526 | 0.21 | 0.42 | 0.00 | 2.28 |
| RES | 10,896 | 0.14 | 0.19 | 0.00 | 3.49 |
| APC | 10,896 | 0.08 | 0.13 | 0.00 | 3.16 |
| G_score | 10,896 | 0.10 | 0.31 | -0.79 | 3.48 |
| C_score | 10,896 | -0.06 | 0.42 | -3.61 | 1.43 |

on CSR will be heavily constrained. Therefore, the negative moderating role of CEO duality is confirmed and H2 is supported.

Turning to institutional investor, see in Table 5, the coefficients of interaction term (InOwner *Coverage) are positively significant (2.368, $p < 0.01$; 0.846, $p < 0.01$; 0.512, $p < 0.01$; 0.441, $p < 0.01$; 0.541, $p < 0.01$) except CSR_Soc (0.029, S.E. = 0.172, n.s). Being long-term oriented, institutional holders intend to support value added programs and CSR-related actions in the long run. With the helping hand lent by institutional investors, the promoting effect of analyst coverage on CSR can be enlarged. Therefore, the positive moderating role of institutional ownership is confirmed and H3 is mainly supported.

As for executive ownership, see in Table 6, the coefficients of interaction term (ExOwner *Coverage) are negatively significant (-1.253, $p < 0.01$; -0.507, $p < 0.01$; -0.148, $p < 0.01$; -0.199, $p < 0.05$; -0.285, $p < 0.01$) except CSR_Soc. (-0.114, S.E. = 0.80, n.s). Higher level of executive ownership usually creates greater incentives and more space for managers to fulfill their own interests while sacrifice the benefits of other stakeholders. While it may be possible for a firm to generate higher current benefits through engaging in less CSR at the expense of giving up future performance in the next periods, managers with larger stock holdings will expropriate resources and reduce social investments. As a result, the monitor of analyst coverage is shrugged off. Hence, the proposed negative moderating effect of executive ownership of H4 is mainly supported.

As for real earnings management, see in Table 7, the coefficients of interaction term (RES *Coverage) are almost positively significant (2.295, $p < 0.05$; 0.449, $p < 0.01$; 0.606, $p < 0.05$) except CSR_Sha, CSR_Emp and CSR_Soc (0.329, S.E. = 0.374, n.s; 0.356, S.E. = 0.278, n.s; 0.556, S.E. = 0.431, n.s). The related firms may cover their less detectable real earnings management and involve more in CSR under financial analyst coverage. Moreover, real earnings management creates greater space for financial analysts to work on CSR. Hence, the proposed

**Table 2. Pearson correlation.**

| | 1 | 2 | 3 | 4 | 5 | 6 | 7 | 8 | 9 | 10 | 11 | 12 | 13 | 14 | 15 | 16 | 17 | 18 | 19 |
|---|---|---|---|---|---|---|---|---|---|---|---|---|---|---|---|---|---|---|---|
| CSR_T | 1 | | | | | | | | | | | | | | | | | | |
| CSR_Sha | 0.595*** | 1 | | | | | | | | | | | | | | | | | |
| CSR_Emp | 0.800*** | 0.163*** | 1 | | | | | | | | | | | | | | | | |
| CSR_SCC | 0.845*** | 0.163*** | 0.844*** | 1 | | | | | | | | | | | | | | | |
| CSR_Env | 0.826*** | 0.138*** | 0.886*** | 0.902*** | 1 | | | | | | | | | | | | | | |
| CSR_Soc | 0.459*** | 0.295*** | 0.098*** | 0.208*** | 0.118*** | 1 | | | | | | | | | | | | | |
| Coverage | 0.341*** | 0.430*** | 0.189*** | 0.181*** | 0.166*** | 0.116*** | 1 | | | | | | | | | | | | |
| Size | 0.194*** | 0.050*** | 0.226*** | 0.190*** | 0.197*** | 0.060*** | 0.330*** | 1 | | | | | | | | | | | |
| PPE | -0.045*** | -0.103*** | -0.011 | 0.026*** | 0.031*** | -0.082*** | -0.110*** | -0.057*** | 1 | | | | | | | | | | |
| Lev | -0.075*** | -0.318*** | 0.090*** | 0.071*** | 0.095*** | -0.059*** | -0.104*** | 0.361*** | 0.173*** | 1 | | | | | | | | | |
| Top10h | 0.125*** | 0.291*** | -0.020** | 0.000 | -0.008 | 0.055*** | 0.212*** | -0.038*** | -0.064*** | -0.243*** | 1 | | | | | | | | |
| Div | 0.250*** | 0.466*** | 0.050*** | 0.054*** | 0.040*** | 0.101*** | 0.339*** | 0.072*** | -0.085*** | -0.262*** | 0.262*** | 1 | | | | | | | |
| TobinQ | -0.001 | 0.169*** | -0.093*** | -0.078*** | -0.100*** | 0.012 | 0.054*** | -0.443*** | -0.045*** | -0.364*** | 0.173*** | 0.059*** | 1 | | | | | | |
| CEODu | -0.040*** | 0.058*** | -0.084*** | -0.073*** | -0.073*** | -0.015 | 0.036*** | -0.147*** | -0.056*** | -0.115*** | 0.113*** | 0.075*** | 0.126*** | 1 | | | | | |
| IntOwner | 0.172*** | 0.059*** | 0.167*** | 0.166*** | 0.145*** | 0.100*** | 0.201*** | 0.359*** | 0.031*** | 0.162*** | 0.083*** | -0.010 | -0.120*** | 0.181*** | 1 | | | | |
| ExOwner | -0.010 | 0.161*** | -0.099*** | -0.085*** | -0.080*** | -0.031*** | 0.076*** | -0.255*** | -0.090*** | -0.212*** | 0.312*** | 0.169*** | 0.210*** | -0.414*** | -0.360*** | 1 | | | |
| RES | -0.038*** | 0.112*** | 0.102*** | 0.169 | 0.039*** | 0.053*** | 0.010 | 0.112*** | -0.023 | -0.079*** | -0.076*** | 0.056*** | 0.056*** | 0.227*** | 0.159*** | 0.074*** | 1 | | |
| APC | 0.083*** | 0.077*** | 0.118*** | 0.051*** | 0.035* | 0.010 | 0.083*** | -0.039** | -0.047** | -0.066*** | 0.040** | 0.025 | 0.169*** | 0.107*** | 0.043** | 0.879*** | 0.012 | 1 | |
| G_score | 0.196*** | -0.037*** | 0.058*** | 0.096*** | 0.151*** | 0.166*** | 0.001 | 0.196*** | -0.037* | 0.058*** | 0.096*** | 0.008 | -0.084*** | 0.032 | -0.141 | 0.091*** | 0.010*** | 0.009 | 1 |
| C_score | -0.034 | -0.151 | -0.064*** | -0.135*** | -0.148*** | -0.158*** | -0.034** | -0.179*** | 0.040** | 0.017*** | -0.105*** | -0.057*** | 0.065*** | -0.129*** | -0.037* | -0.028 | -0.753*** | 0.012*** | -0.029** |

$p < 0.1$,

** $p < 0.05$,

*** $p < 0.01$

**Table 3. Effect of analyst coverage on CSR.**

|  | CSR_T | CSR_Sha | CSR_Emp | CSR_SCC | CSR_Env | CSR_Soc |
|---|---|---|---|---|---|---|
| Constant | 14.283 | 39.500*** | -7.328*** | -10.847*** | -10.740*** | 3.699 |
|  | (13.073) | (5.232) | (2.437) | (3.575) | (4.142) | (3.490) |
| Coverage | 1.410*** | 0.997*** | 0.110*** | 0.129** | 0.141** | 0.033 |
|  | (0.212) | (0.084) | (0.038) | (0.060) | (0.069) | (0.051) |
| Size | 0.476 | -1.475*** | 0.522*** | 0.686*** | 0.733*** | 0.010 |
|  | (0.619) | (0.247) | (0.116) | (0.169) | (0.198) | (0.166) |
| PPE | -1.925 | -3.150*** | 0.173 | 0.679 | 0.559 | -0.186 |
|  | (3.036) | (1.161) | (0.564) | (0.893) | (0.993) | (0.584) |
| Lev | 0.547 | -2.468*** | 0.901*** | 1.184** | 1.158** | -0.228 |
|  | (1.750) | (0.690) | (0.314) | (0.502) | (0.585) | (0.406) |
| Top10h | -0.010 | 0.072*** | -0.018*** | -0.028*** | -0.040*** | 0.004 |
|  | (0.025) | (0.009) | (0.004) | (0.007) | (0.008) | (0.006) |
| Div | 1.229*** | 0.959*** | 0.033 | 0.059 | 0.065 | 0.114 |
|  | (0.447) | (0.168) | (0.081) | (0.125) | (0.143) | (0.125) |
| TobinQ | 0.476*** | 0.140** | 0.049** | 0.104** | 0.131*** | 0.052 |
|  | (0.130) | (0.056) | (0.023) | (0.041) | (0.043) | (0.033) |
| Year | Control | Control | Control | Control | Control | Control |
| Firm fixed | Control | Control | Control | Control | Control | Control |
| N | 10896 | 10896 | 10896 | 10896 | 10896 | 10896 |
| Adj. $R^2$ | 0.140 | 0.095 | 0.122 | 0.137 | 0.132 | 0.012 |

Standard errors in parentheses

* $p < 0.1$,

** $p < 0.05$,

*** $p < 0.01$

positive moderating effect of real earnings management of H5 is mainly supported. When we focus real earnings management on enlarging production to adjust producing cost, the coefficients of interaction term (*APC* *Coverage*) are also positively significant (2.823, $p < 0.05$; 0.686, $p < 0.01$; 0.803, $p < 0.05$; 0.658, $p < 0.05$) except *CSR_Sha* and *CSR_Soc* (-0.040, S.E. = 0.501, n.s; 0.716, S.E. = 0.579, n.s), as shown in Table 8.

As for accounting conservatism, see in Table 9, the coefficients of interaction term (*C_score* *Coverage*) are negatively significant (-1.363, S.E. = 0.469, $p < 0.01$; -0.213, S.E. = 0.099, $p < 0.05$; -0.493, S.E. = 0.134, $p < 0.01$; -0.488, S.E. = 0.171, $p < 0.01$; -0.222, S.E. = 0.093, $p < 0.05$) except *CSR_Sha* (0.052, S.E. = 0.131, n.s). The related firms, with a high degree of accounting conservatism, can consist the behaviors of the insiders in line with the outsiders by providing more conservative accounting information. Hence, accounting conservatism reduce the effect of analyst coverage on CSR, and H6 is supported. When we view the opposite of accounting conservatism, the conclusion is similar, see in Table 10. *G_score* stands for the timeliness of good news. The more rapid response to good news, the less accounting conservatism it is and the larger *G_score* will be. The coefficients of interaction term *G_score* *Coverage*) are mostly positive significant (1.761, S.E. = 0.704, $p < 0.05$; 0.306, S.E. = 0.156, $p < 0.1$; 0.796, S.E. = 0.200, $p < 0.01$; 0.804, S.E. = 0.263, $p < 0.01$) except *CSR_Sha* and *CSR_Soc* (-0.339, S.E. = 0.184, $p < 0.1$; 0.195, S.E. = 0.134, n.s).

**Table 4. The moderating role of managers' duality.**

|  | CSR_T | CSR_Sha | CSR_Emp | CSR_SCC | CSR_Env | CSR_Soc |
|---|---|---|---|---|---|---|
| Constant | 12.356 | 38.343*** | -7.633*** | -10.925*** | -11.032*** | 3.603 |
|  | (13.130) | (5.262) | (2.428) | (3.620) | (4.167) | (3.546) |
| Coverage | 1.396*** | 0.993*** | 0.108*** | 0.127** | 0.136** | 0.032 |
|  | (0.212) | (0.085) | (0.038) | (0.060) | (0.069) | (0.051) |
| CEODu | -0.540 | 0.179 | -0.154* | -0.300* | -0.254 | -0.011 |
|  | (0.532) | (0.236) | (0.089) | (0.154) | (0.165) | (0.140) |
| CEODu *Coverage | -1.379*** | -0.237* | -0.185*** | -0.289*** | -0.361*** | -0.308*** |
|  | (0.360) | (0.143) | (0.064) | (0.110) | (0.115) | (0.087) |
| Size | 0.568 | -1.422*** | 0.538*** | 0.692*** | 0.749*** | 0.012 |
|  | (0.622) | (0.248) | (0.116) | (0.171) | (0.200) | (0.168) |
| PPE | -1.698 | -3.035*** | 0.238 | 0.697 | 0.615 | -0.212 |
|  | (3.065) | (1.169) | (0.566) | (0.904) | (1.000) | (0.590) |
| Lev | 0.710 | -2.448*** | 0.933*** | 1.253** | 1.196** | -0.224 |
|  | (1.749) | (0.692) | (0.314) | (0.503) | (0.585) | (0.405) |
| Top10h | -0.006 | 0.073*** | -0.018*** | -0.027*** | -0.039*** | 0.005 |
|  | (0.025) | (0.009) | (0.004) | (0.007) | (0.008) | (0.006) |
| Div | 1.135** | 0.938*** | 0.020 | 0.051 | 0.035 | 0.091 |
|  | (0.447) | (0.171) | (0.081) | (0.125) | (0.144) | (0.126) |
| TobinQ | 0.493*** | 0.141** | 0.050** | 0.110*** | 0.136*** | 0.057* |
|  | (0.131) | (0.056) | (0.023) | (0.041) | (0.043) | (0.033) |
| Year | Control | Control | Control | Control | Control | Control |
| Firm fixed | Control | Control | Control | Control | Control | Control |
| N | 10820 | 10820 | 10820 | 10820 | 10820 | 10820 |
| Adj. $R^2$ | 0.143 | 0.095 | 0.124 | 0.139 | 0.134 | 0.013 |

Standard errors in parentheses

* $p < 0.1$,

** $p < 0.05$,

*** $p < 0.01$

## Robustness check

The previous results indicate that analyst coverage would exert significant influence on firms' CSR performance. However, it should be noticed that the reverse logic might exist because firms with better CSR performance could attract greater attention from analysts in advance [16]. Therefore, an instrument variable for analyst coverage is constructed. In addition, 2SLS regression is conducted to mitigate the concern of endogeneity. Our instrument, expected coverage (*ExpCoverage*), is applied to proxy the evolvement of brokerage institution size that is closely linked to the operating plans and performances of the organizations [22, 41]. While the size of outside brokerage institution is unlikely depended on a firm's inside decision of CSR that the analysts pay attention to, hence, the difference of analyst coverage triggered by the expand or contract of brokerage institutions size could enhance the causality by mitigating the endogeneity concern.

Expected coverage is calculated by using the following equation:

$$ExpCoverage_{i,t} = \sum_{j=1}^{n} (Brokersize_{t,j}/Brokersize_{0,j}) \times Coverage_{i,0,j} \qquad (18)$$

**Table 5. The moderating role of institutional ownership.**

|  | CSR_T | CSR_Sha | CSR_Emp | CSR_SCC | CSR_Env | CSR_Soc |
|---|---|---|---|---|---|---|
| Constant | 15.687 | 40.390*** | -7.441*** | -10.715*** | -10.763*** | 4.218 |
|  | (13.109) | (5.232) | (2.442) | (3.567) | (4.167) | (3.497) |
| Coverage | 1.262*** | 0.966*** | 0.083** | 0.089 | 0.100 | 0.023 |
|  | (0.207) | (0.084) | (0.037) | (0.058) | (0.067) | (0.051) |
| InOwner | 2.981*** | -0.652 | 0.643*** | 1.376*** | 1.495*** | 0.118 |
|  | (1.069) | (0.442) | (0.193) | (0.294) | (0.346) | (0.235) |
| InOwner*Coverage | 2.368*** | 0.846*** | 0.512*** | 0.441** | 0.541** | 0.029 |
|  | (0.751) | (0.269) | (0.143) | (0.219) | (0.252) | (0.172) |
| Size | 0.423 | -1.520*** | 0.530*** | 0.687*** | 0.740*** | -0.015 |
|  | (0.621) | (0.246) | (0.116) | (0.169) | (0.200) | (0.166) |
| PPE | -2.407 | -3.233*** | 0.116 | 0.494 | 0.416 | -0.201 |
|  | (3.051) | (1.166) | (0.572) | (0.894) | (1.002) | (0.588) |
| Lev | 0.016 | -2.320*** | 0.724** | 0.905* | 0.891 | -0.183 |
|  | (1.759) | (0.699) | (0.316) | (0.499) | (0.595) | (0.407) |
| Top10h | -0.015 | 0.076*** | -0.019*** | -0.031*** | -0.044*** | 0.003 |
|  | (0.025) | (0.009) | (0.005) | (0.007) | (0.008) | (0.006) |
| Div | 1.201*** | 0.975*** | 0.017 | 0.036 | 0.039 | 0.135 |
|  | (0.451) | (0.171) | (0.081) | (0.126) | (0.145) | (0.126) |
| TobinQ | 0.338*** | 0.138** | 0.022 | 0.049 | 0.083* | 0.047 |
|  | (0.130) | (0.057) | (0.023) | (0.039) | (0.043) | (0.034) |
| Year | Control | Control | Control | Control | Control | Control |
| Firm fixed | Control | Control | Control | Control | Control | Control |
| N | 10724 | 10724 | 10724 | 10724 | 10724 | 10724 |
| Adj. $R^2$ | 0.140 | 0.095 | 0.123 | 0.137 | 0.132 | 0.011 |

Standard errors in parentheses

* $p < 0.1$,

** $p < 0.05$,

*** $p < 0.01$

In Eq (18), $Coverage_{i,0,j}$ equals to the number of analysts, employed by broke $r$ $j$, who broadcasted firm $i$, while $0$ signals it is the benchmark year. $Brokersize_{0,j}$ is the revenue of broker $j$ belongs to benchmark fiscal year of $0$. $Brokersize_{t,j}$ equals to the revenue of broker $j$ earned in the fiscal year of $t$. $ExpCoverage_{i,t}$ shows the expected analyst coverage of firm $i$ in the fiscal year of $t$.

The regression results of 2SLS analysis are shown in Table 11. The first column shows the results of the first-stage regression. *Coverage* is the dependent variable. *ExpCoverage* is the independent variable, and is calculated by the logarithm of expected coverage plus one. The estimated coefficient of *ExpCoverage* is 0.180 and significant at the level of 1% (S.E. = 0.016). In line with the extant literature [14, 22], *ExpCoverage* is significantly linked to *Coverage*. Thus, expected analyst coverage is a potent instrument variable of the original analyst coverage. The null hypothesis is not accepted. The coefficients estimated in the second stage are probably not biased and the conclusions are considerably reliable.

Columns 2 to 7 report the results of second-stage regression with results of *CSR_T*, *CSR_Sha*, *CSR_Emp*, *CSR_SCC*, *CSR_Env*, and *CSR_Soc* as the dependent variables respectively. The coefficients of *Coverage*, as the independent variable, are almost positive and

**Table 6. The moderating role of executive ownership.**

|  | CSR_T | CSR_Sha | CSR_Emp | CSR_SCC | CSR_Env | CSR_Soc |
|---|---|---|---|---|---|---|
| Constant | 18.182 | 40.209*** | -7.131*** | -9.712*** | -9.857** | 4.673 |
|  | (13.429) | (5.256) | (2.491) | (3.707) | (4.293) | (3.513) |
| Coverage | 1.493*** | 1.005*** | 0.124*** | 0.152** | 0.171** | 0.041 |
|  | (0.220) | (0.086) | (0.040) | (0.062) | (0.071) | (0.052) |
| ExOwner | -0.777 | 1.099*** | -0.383*** | -0.678*** | -0.690*** | -0.125 |
|  | (0.575) | (0.290) | (0.102) | (0.156) | (0.180) | (0.142) |
| ExOwner *Coverage | -1.253*** | -0.507*** | -0.148*** | -0.199** | -0.285*** | -0.114 |
|  | (0.325) | (0.169) | (0.055) | (0.085) | (0.099) | (0.080) |
| Size | 0.295 | -1.509*** | 0.514*** | 0.633*** | 0.694*** | -0.038 |
|  | (0.636) | (0.248) | (0.118) | (0.176) | (0.206) | (0.168) |
| PPE | -2.666 | -2.823** | 0.014 | 0.312 | 0.171 | -0.339 |
|  | (3.039) | (1.175) | (0.574) | (0.913) | (1.001) | (0.596) |
| Lev | 0.030 | -2.311*** | 0.713** | 0.924* | 0.832 | -0.127 |
|  | (1.783) | (0.699) | (0.319) | (0.511) | (0.596) | (0.419) |
| Top10h | -0.001 | 0.066*** | -0.015*** | -0.022*** | -0.035*** | 0.005 |
|  | (0.025) | (0.009) | (0.005) | (0.007) | (0.008) | (0.006) |
| Div | 1.216*** | 0.915*** | 0.033 | 0.066 | 0.087 | 0.115 |
|  | (0.456) | (0.171) | (0.083) | (0.128) | (0.148) | (0.128) |
| TobinQ | 0.443*** | 0.148*** | 0.043* | 0.092** | 0.121*** | 0.040 |
|  | (0.134) | (0.055) | (0.024) | (0.042) | (0.044) | (0.034) |
| Year | Control | Control | Control | Control | Control | Control |
| Firm fixed | Control | Control | Control | Control | Control | Control |
| N | 10526 | 10526 | 10526 | 10526 | 10526 | 10526 |
| Adj. $R^2$ | 0.143 | 0.098 | 0.125 | 0.138 | 0.134 | 0.011 |

Standard errors in parentheses

* $p < 0.1$,

** $p < 0.05$,

*** $p < 0.01$

significant (13.177, S.E. = 1.803, $p < 0.01$; 4.050, S.E. = 0.562, $p < 0.01$; 3.440, S.E. = 0.555, $p < 0.01$; 3.156, S.E. = 0.595, $p < 0.01$; 0.298, S.E. = 0.357, n.s.), enhancing the prior conclusions that analyst coverage improves CSR performance. Referring to the former results, an interesting phenomenon is observed that the magnitudes of the estimated coefficients of the 2SLS are fairly larger than the OLS. Accordingly, we can get that the coefficients of regressions are downwardly biased resulting by the problem of endogeneity. This conclusion implies that certain factors, which can both induce the firms less socially responsible and reduce the attention of analysts, are ignored. By mitigating the endogeneity problem of omitting variable, the effect of analysts on corporate socially benefit behaviors gets more positive. Therefore, we can get that the causal relationship between analysts and CSR are probably exist, by applying the 2SLS regression.

## Discussion and conclusion

### Conclusions

CSR has become a focus of attention both in the field of academics and practice all over the world, especially in the emerging markets such as China [44–51]. Following this research of

**Table 7. The moderating role of real earnings management.**

|  | CSR_T | CSR_Sha | CSR_Emp | CSR_SCC | CSR_Env | CSR_Soc |
|---|---|---|---|---|---|---|
| Constant | 12.924 | 40.207*** | -9.552*** | -11.715** | -10.970** | 4.954 |
|  | (16.545) | (6.001) | (3.097) | (4.742) | (5.472) | (4.317) |
| Coverage | 0.526 | 1.129** | 0.115 | -0.206 | -0.366 | -0.145 |
|  | (1.193) | (0.456) | (0.205) | (0.325) | (0.353) | (0.452) |
| RES | 1.571*** | 1.004*** | 0.149*** | 0.178** | 0.211** | 0.028 |
|  | (0.266) | (0.102) | (0.048) | (0.075) | (0.086) | (0.067) |
| RES*Coverage | 2.295** | 0.329 | 0.449*** | 0.606** | 0.356 | 0.556 |
|  | (1.025) | (0.374) | (0.163) | (0.290) | (0.278) | (0.431) |
| Size | 0.626 | -1.531*** | 0.649*** | 0.761*** | 0.788*** | -0.041 |
|  | (0.773) | (0.282) | (0.145) | (0.222) | (0.258) | (0.200) |
| PPE | -2.603 | -0.101 | -0.476 | -0.614 | -1.161 | -0.252 |
|  | (3.934) | (1.372) | (0.743) | (1.175) | (1.322) | (0.681) |
| Lev | -0.302 | -2.685*** | 0.654* | 1.025* | 0.994 | -0.290 |
|  | (2.110) | (0.781) | (0.392) | (0.622) | (0.728) | (0.449) |
| Top10h | 0.001 | 0.067*** | -0.014** | -0.021** | -0.034*** | 0.003 |
|  | (0.030) | (0.010) | (0.006) | (0.009) | (0.011) | (0.007) |
| Div | 1.079** | 0.843*** | 0.053 | 0.028 | 0.061 | 0.094 |
|  | (0.499) | (0.187) | (0.091) | (0.140) | (0.164) | (0.143) |
| TobinQ | 0.394** | 0.185*** | -0.000 | 0.061 | 0.066 | 0.083* |
|  | (0.171) | (0.069) | (0.031) | (0.054) | (0.057) | (0.045) |
| Year | Control | Control | Control | Control | Control | Control |
| Firm fixed | Control | Control | Control | Control | Control | Control |
| N | 8044 | 8044 | 8044 | 8044 | 8044 | 8044 |
| Adj. $R^2$ | 0.164 | 0.078 | 0.159 | 0.172 | 0.165 | 0.015 |

Standard errors in parentheses

* $p < 0.1$,

** $p < 0.05$,

*** $p < 0.01$

the impact of analysts on CSR, this study theoretically and empirically examines whether analysts, an external monitoring stakeholder, care about CSR issues. Based upon Chinese listed firms in the manufacturing industry, the robust empirical results show that analyst coverage significantly promotes firm's CSR. It is also observed that high institutional ownership could strengthen the margin contribution of analyst coverage on CSR while CEO duality and executive ownership act as negative moderators. High real earnings management degree and less accounting conservatism enlarge the promoting role of financial coverage on CSR. It is worth to be noticed that the governance characters and information characters do not work in the same way. The improvements in corporate governance, stemming from analysts' monitoring function, play complementary roles in giving impetus to socially responsible investments with long horizons. The weak information condition makes more demand for the analyst coverage to keep corporations engaged in social benefit activities.

## Theoretical implications

This research also provides implication for the theoretical and practical understanding of analyst coverage's role on CSR.

**Table 8. The moderating role of abnormal production cost.**

|  | CSR_T | CSR_Sha | CSR_Emp | CSR_SCC | CSR_Env | CSR_Soc |
|---|---|---|---|---|---|---|
| Constant | 13.307 | 40.430*** | -9.515*** | -11.669** | -10.878** | 4.939 |
|  | (16.562) | (6.021) | (3.094) | (4.734) | (5.453) | (4.344) |
| Coverage | 1.558*** | 1.002*** | 0.148*** | 0.174** | 0.210** | 0.024 |
|  | (0.266) | (0.102) | (0.048) | (0.075) | (0.086) | (0.066) |
| APC | 1.008 | 1.900*** | 0.155 | -0.317 | -0.392 | -0.338 |
|  | (1.503) | (0.620) | (0.266) | (0.400) | (0.464) | (0.643) |
| APC*Coverage | 2.823** | -0.040 | 0.686*** | 0.803** | 0.658** | 0.716 |
|  | (1.199) | (0.501) | (0.206) | (0.345) | (0.329) | (0.579) |
| Size | 0.614 | -1.540*** | 0.649*** | 0.760*** | 0.785*** | -0.039 |
|  | (0.773) | (0.283) | (0.145) | (0.222) | (0.257) | (0.202) |
| PPE | -2.689 | -0.153 | -0.494 | -0.623 | -1.154 | -0.264 |
|  | (3.934) | (1.372) | (0.743) | (1.176) | (1.322) | (0.681) |
| Lev | -0.307 | -2.653*** | 0.648* | 1.016 | 0.979 | -0.297 |
|  | (2.112) | (0.782) | (0.392) | (0.622) | (0.728) | (0.449) |
| Top10h | 0.000 | 0.067*** | -0.014** | -0.021** | -0.034*** | 0.003 |
|  | (0.030) | (0.010) | (0.006) | (0.009) | (0.011) | (0.007) |
| Div | 1.091** | 0.853*** | 0.055 | 0.028 | 0.061 | 0.094 |
|  | (0.499) | (0.187) | (0.091) | (0.140) | (0.164) | (0.143) |
| TobinQ | 0.400** | 0.187*** | 0.000 | 0.063 | 0.064 | 0.085* |
|  | (0.171) | (0.069) | (0.031) | (0.054) | (0.057) | (0.046) |
| Year | Control | Control | Control | Control | Control | Control |
| Firm fixed | Control | Control | Control | Control | Control | Control |
| N | 8044 | 8044 | 8044 | 8044 | 8044 | 8044 |
| Adj. $R^2$ | 0.164 | 0.078 | 0.160 | 0.172 | 0.165 | 0.015 |

Standard errors in parentheses

* $p < 0.1$,

** $p < 0.05$,

*** $p < 0.01$

First, this research enriches the study on the internal channels, constraint conditions, or mechanisms, of analysts working on CSR. Specifically, this research identifies five contingent factors, specifically CEO duality, institutional ownership, executive ownership, real earnings management, and accounting conservatism, and confirming the conclusion that the combination among inside and outside factors can make a larger margin contribution to CSR. This research enriches the prior literature by disclosing the different roles of financial analyst on CSR, by the governance channel and the information channel. On the one side, analyst coverage and firms' inside governance mechanism complements with each other in the field of CSR. The impact of financial analyst on CSR is weaker in an inferior inside governance condition. On the other side, analyst coverage substitutes with information condition in the aspect of impacting CSR. The positive role of analyst coverage on CSR is more obvious in the worse information condition.

To be specifically, the effect of analyst coverage depends on the synergy of various factors, and is uniquely contingent on interior motivations to achieve substantive CSR. The mitigation in agency problem and information asymmetry from analysts' monitoring and the improvements in corporate governance play complementary roles in giving impetus to socially

**Table 9. The moderating role of accounting conservatism.**

|  | CSR_T | CSR_Sha | CSR_Emp | CSR_SCC | CSR_Env | CSR_Soc |
|---|---|---|---|---|---|---|
| Constant | 2.594 | 31.996*** | -9.867*** | -12.141** | -11.217* | 3.823 |
|  | (18.207) | (6.443) | (3.503) | (5.374) | (6.191) | (4.446) |
| Coverage | 1.604*** | 0.953*** | 0.156*** | 0.189** | 0.223** | 0.083 |
|  | (0.286) | (0.104) | (0.052) | (0.083) | (0.096) | (0.062) |
| C_score | -1.261** | 0.024 | -0.284*** | -0.464*** | -0.617*** | 0.080 |
|  | (0.512) | (0.138) | (0.105) | (0.146) | (0.190) | (0.099) |
| C_score *Coverage | -1.363*** | 0.052 | -0.213** | -0.493*** | -0.488*** | -0.222** |
|  | (0.469) | (0.131) | (0.099) | (0.134) | (0.171) | (0.093) |
| Size | 1.172 | -1.144*** | 0.670*** | 0.805*** | 0.813*** | 0.027 |
|  | (0.862) | (0.306) | (0.165) | (0.254) | (0.294) | (0.209) |
| PPE | -2.078 | -0.643 | -0.406 | -0.315 | -0.573 | -0.141 |
|  | (3.825) | (1.348) | (0.745) | (1.178) | (1.334) | (0.706) |
| Lev | 0.866 | -2.152** | 0.916** | 1.210* | 1.345 | -0.452 |
|  | (2.407) | (0.873) | (0.443) | (0.723) | (0.829) | (0.508) |
| Top10h | -0.034 | 0.078*** | -0.021*** | -0.038*** | -0.050*** | -0.003 |
|  | (0.031) | (0.011) | (0.006) | (0.010) | (0.011) | (0.007) |
| Div | 1.117* | 1.186*** | -0.035 | -0.032 | -0.026 | 0.025 |
|  | (0.575) | (0.208) | (0.103) | (0.158) | (0.191) | (0.161) |
| TobinQ | 0.638*** | 0.147* | 0.055 | 0.170*** | 0.171*** | 0.095** |
|  | (0.194) | (0.082) | (0.036) | (0.063) | (0.065) | (0.044) |
| Year | Control | Control | Control | Control | Control | Control |
| Firm fixed | Control | Control | Control | Control | Control | Control |
| N | 7230 | 7230 | 7230 | 7230 | 7230 | 7230 |
| Adj. $R^2$ | 0.169 | 0.084 | 0.157 | 0.172 | 0.164 | 0.018 |

Standard errors in parentheses

* $p < 0.1$,

** $p < 0.05$,

*** $p < 0.01$

responsible investments with long horizons in the governance field, while acting as a substitute function in the information area. For example, the separation of CEO and chair and the participation of institutional investors are encouraged, and stock-based compensation to managers or directors needs care. The worse information situations demand analyst coverage to shape firms socially benefit behaviors more. For example, it shows that analyst coverage makes a deeper effect on CSR, in the worse information condition, specifically higher real earnings management and less accounting conservatism condition. All in all, financial analysts play different role with governance and information context. Analyst coverage complements with governance system, while substitutes for information channels to make a difference in corporate social friend activities.

Second, this study deepens the understanding of CSR practices, especially in emerging economies. Evidence has shown that CSR of firms in China are significantly impacted by informal institutions [12]. While the undeveloped environment of information, analysts could fill the information gap and confine the agency problem. If analyst recommendations contain additional information content related to socially responsible issues, idiosyncratic CSR strategies are a reasonable candidate as right response to a great diversity of stakeholder demands

Table 10. The moderating role of good news timeliness in earnings.

|  | CSR_T | CSR_Sha | CSR_Emp | CSR_SCC | CSR_Env | CSR_Soc |
|---|---|---|---|---|---|---|
| Constant | 2.177 | 32.003*** | -10.135*** | -11.681** | -11.720* | 3.710 |
|  | (18.205) | (6.436) | (3.497) | (5.355) | (6.169) | (4.470) |
| Coverage | 1.610*** | 0.950*** | 0.155*** | 0.189** | 0.220** | 0.095 |
|  | (0.283) | (0.104) | (0.052) | (0.082) | (0.095) | (0.062) |
| G_score | 2.065*** | -0.037 | 0.515*** | 0.871*** | 0.991*** | -0.275* |
|  | (0.772) | (0.211) | (0.161) | (0.230) | (0.294) | (0.156) |
| G_score *Coverage | 1.761** | -0.339* | 0.306* | 0.796*** | 0.804*** | 0.195 |
|  | (0.704) | (0.184) | (0.156) | (0.200) | (0.263) | (0.134) |
| Size | 1.203 | -1.143*** | 0.685*** | 0.787*** | 0.840*** | 0.033 |
|  | (0.862) | (0.305) | (0.165) | (0.253) | (0.294) | (0.209) |
| PPE | -2.097 | -0.864 | -0.278 | -0.259 | -0.543 | -0.154 |
|  | (3.813) | (1.340) | (0.747) | (1.173) | (1.323) | (0.702) |
| Lev | 0.320 | -2.312*** | 0.799* | 1.065 | 1.178 | -0.410 |
|  | (2.408) | (0.882) | (0.447) | (0.718) | (0.832) | (0.511) |
| Top10h | -0.033 | 0.079*** | -0.021*** | -0.038*** | -0.050*** | -0.003 |
|  | (0.031) | (0.011) | (0.006) | (0.009) | (0.011) | (0.007) |
| Div | 1.103* | 1.124*** | -0.023 | -0.017 | 0.007 | 0.012 |
|  | (0.574) | (0.207) | (0.102) | (0.158) | (0.190) | (0.160) |
| TobinQ | 0.640*** | 0.148* | 0.058 | 0.174*** | 0.175*** | 0.085* |
|  | (0.194) | (0.082) | (0.036) | (0.063) | (0.065) | (0.044) |
| Year | Control | Control | Control | Control | Control | Control |
| Firm fixed | Control | Control | Control | Control | Control | Control |
| N | 7230 | 7230 | 7230 | 7230 | 7230 | 7230 |
| Adj. $R^2$ | 0.168 | 0.083 | 0.158 | 0.174 | 0.166 | 0.017 |

Standard errors in parentheses

* $p < 0.1$,

** $p < 0.05$,

*** $p < 0.01$

and expectations [6]. Thus, firms are inclined to adjust their CSR practices corresponding to the distinct levels of supervision that managers face. For the ongoing progress and quick expansion of economic growth, intensified public concern and supervision are imperative to simultaneously boost the well-being of wider society.

## Practical implications

Meanwhile, this study also provides practical implications for governments and firms.

Frist, the concentration power of top managers should be alerted, in the way of harming CSR. According to our results, the function of financial analysts on CSR significantly decreases in the firms with strong or concentrated managers' power. The managers should cooperate with the financial analysts to enhance the positive effect of analysts' coverage on CSR. To the opposite, when the managers are equipped with negotiated power, they will less care for CSR even the firms are followed by more financial analysts.

Second, it is noticeable that financial analysts should give more attention to the firms with low accounting information quality, in order to keep firm investing towards CSR. In our paper, we find that financial analysts promote CSR to a larger degree in the firms with low

**Table 11. Two-stage least squares (2SLS) regression with the instrument of expected analyst coverage.**

| | First-stage Coverage | Second-stage | | | | | |
|---|---|---|---|---|---|---|---|
| | | CSR_T | CSR_Sha | CSR_Emp | CSR_SCC | CSR_Env | CSR_Soc |
| Constant | -11.33*** | 55.926** | 20.813*** | 10.828** | 15.541** | 7.980 | 0.764 |
| | (0.453) | (25.207) | (7.874) | (5.190) | (7.781) | (8.389) | (4.917) |
| ExpCoverage | 0.180*** | | | | | | |
| | (0.016) | | | | | | |
| Coverage | | 13.177*** | 4.050*** | 2.234*** | 3.440*** | 3.156*** | 0.298 |
| | | (1.803) | (0.562) | (0.371) | (0.555) | (0.595) | (0.357) |
| Size | 0.563*** | -2.232* | -0.815** | -0.501* | -0.730* | -0.365 | 0.179 |
| | (0.021) | (1.268) | (0.393) | (0.261) | (0.390) | (0.421) | (0.249) |
| PPE | -0.445*** | -3.887 | -0.501 | -1.633*** | 0.877 | -0.654 | -1.976*** |
| | (0.133) | (3.007) | (0.922) | (0.591) | (0.957) | (1.048) | (0.552) |
| Lev | -0.547*** | -3.195 | -4.557*** | 0.782* | 0.578 | 0.801 | -0.799* |
| | (0.089) | (2.217) | (0.657) | (0.469) | (0.706) | (0.751) | (0.434) |
| Top10h | 0.003** | 0.074** | 0.015* | 0.010 | 0.015 | 0.010 | 0.023*** |
| | (0.001) | (0.030) | (0.008) | (0.006) | (0.010) | (0.010) | (0.005) |
| Div | 0.615*** | 1.297 | 2.898*** | -0.543* | -1.004** | -0.736 | 0.680** |
| | (0.037) | (1.483) | (0.454) | (0.300) | (0.445) | (0.492) | (0.282) |
| TobinQ | 0.185*** | -0.915** | 0.098 | -0.285*** | -0.348** | -0.387** | 0.007 |
| | (0.013) | (0.458) | (0.138) | (0.090) | (0.147) | (0.151) | (0.089) |
| Year | Control | Control | Control | Control | Control | Control | Control |
| Firm fixed | Control | Control | Control | Control | Control | Control | Control |
| N | 3,051 | 3051 | 3051 | 3051 | 3051 | 3051 | 3051 |
| Adj. $R^2$ | 0.497 | 0.224 | 0.422 | 0.121 | 0.121 | 0.176 | 0.203 |

Standard errors in parentheses

* $p < 0.1$,

** $p < 0.05$,

*** $p < 0.01$

accounting quality. Considering the role of financial analysts in the worse accounting information condition, both the government and stakeholders should value the function of financial analysts in the aspect of CSR.

Third, policy makers should place greater emphasis on designing appropriate policies or institutional arrangements to give impetus to corporate social responsible activities. In particular, policies need to consider the positive externalities of financial analysts and motivate firms to realize promising financial benefits through CSR enforcement. The China Securities Regulatory Commission (CSRC) should go a step further in formulating the disclosure of CSR, establishing a reward and punishment system, and encouraging the participation of long-term institutional investors. Unobstructed and effective channels for expressing public opinion are also valuable supplements to improving social welfare. Certainly, more efforts should be put to recognize a series of remarkable incentive- or monitoring-based governance tools to prevent analysts from exerting earning pressures on managers [22].

## Limitations and future research directions

Several limitations need to be explicitly notified for the potential readers and future research. First, the CSR measure used in the study is HEXUN CSR Ranking. Being a multidimensional

concept, it is challengeable to construct a comprehensive evaluation of CSR. Future related study could solve this crucial issue by exploiting a more reliable method to measure the core and all dimensions of corporate socially responsible performance. Second, the analyst coverage measure used in this study is the total number of financial analyst teams reporting a firm. It is meaningful to apply alternative measures, e.g. distinguishing star and ordinary analysts, to reinforce the objectivity and reliability of findings. Third, the descriptive results only present the situation of China in the last decade. As such, these findings are likely not smoothly generalizable to other countries or historical contexts.

Overall, this research has made a small contribution but meaningful and vital supplement of study on CSR focusing on financial analysts in China. Analyst coverage complements with governance mechanism and substitutes for information mechanism to work on CSR. It hopefully provides new evidence on the deeper understanding of the analyst-based mechanism underlying the promotions of CSR.

## Supporting information

**S1 Data.**
(ZIP)

## Acknowledgments

We thank helpful comments from the Editor and anonymous reviewers.

## Author Contributions

**Conceptualization:** Xiuhong Du.

**Data curation:** Yan Liu, Xiuhong Du.

**Investigation:** Yan Liu.

**Methodology:** Yan Liu, Xiuhong Du.

**Writing – original draft:** Yan Liu.

**Writing – review & editing:** Yan Liu, Xiuhong Du.

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
