## [Decision Letter · Decision Letter 0]

24 Jul 2023

PONE-D-23-17772How does analyst coverage influence corporate social responsibility (CSR)? The information-based mechanismPLOS ONE

Dear Dr. Du,

Thank you for submitting your manuscript to PLOS ONE. After careful consideration, we feel that it has merit but does not fully meet PLOS ONE’s publication criteria as it currently stands. Therefore, we invite you to submit a revised version of the manuscript that addresses the points raised during the review process.

We look forward to receiving your revised manuscript.

Kind regards,

Yuan-Teng Hsu, Ph.D.

Academic Editor

PLOS ONE

Journal Requirements:

5. We note you have included a table to which you do not refer in the text of your manuscript. Please ensure that you refer to Table 9 in your text; if accepted, production will need this reference to link the reader to the Table.

Additional Editor Comments:

This manuscript was read by two expert reviewers and me. The two reviewers provided excellent feedback. The authors may further enhance the theoretical contributions and robustness of analyses by addressing the comments from the review team.

Reviewers' comments:

Reviewer's Responses to Questions

**Comments to the Author**

1. Is the manuscript technically sound, and do the data support the conclusions?

Reviewer #1: Yes

Reviewer #2: Yes

2. Has the statistical analysis been performed appropriately and rigorously? 

Reviewer #1: Yes

Reviewer #2: Yes

3. Have the authors made all data underlying the findings in their manuscript fully available?

Reviewer #1: Yes

Reviewer #2: Yes

4. Is the manuscript presented in an intelligible fashion and written in standard English?

Reviewer #1: Yes

Reviewer #2: Yes

5. Review Comments to the Author

Reviewer #1: This paper documents the positive association between analyst coverage and CSR. Authors demonstrate that both governance channel and information channel play important role in the positive association. The main contribution of this article is to document the importance of information channel.

I enjoyed reading this paper. CSR is a trendy topic, but its economic mechanism is not well studied. I also appreciate various tests authors did to support their hypotheses. However, I have some concerns on economic mechanism and the execution of the tests.

Major concerns:

1) The authors could expand on the theoretical mechanisms linking analyst coverage to CSR. In particular, how does information asymmetry moderate the relationship between coverage and CSR engagement?As authors explained, the key economic mechanism, information function, refers to ‘reliable degree of external financial information’; if firms are covered by more analyst teams, investors will pay more attention to firms’ CSR level. Does it necessarily lead to monotonical increase/decrease in CSR? Some specific examples of how reduced information asymmetry impacts CSR spending would help readers better understand the posited mechanisms. Currently, the discussion focuses more on the link between asymmetry and stock prices. Clarifying the CSR mechanism specifically would make an important contribution.

2) The monitoring and information channels seem potentially overlapping - are they mutually exclusive? Given that gauging the information channel is a key contribution, it would be helpful for the authors to further distinguish it from the monitoring channel and quantify each one's relative contribution to the empirical results. For instance, the moderating effects of earnings news and accounting conservatism are intended to provide evidence of the information channel, but may also reflect monitoring. Discussing the degree to which the findings uniquely support the information channel would further strengthen the paper's contribution.

3) As authors mentioned, the result could be driven by reverse causality. Authors employ 2SLS to address endogeneity, however, there are several additional tests that could be done before authors claim causality. For example, i) industry*year fixed effect in case there is any shift in the distribution of CSR ii) matching tests given endogeneity of analyst coverage (like PSM).

Minor issues:

Cross reference: Su (2019) is not cited in reference.

Reviewer #2: How does analyst coverage influence corporate social responsibility (CSR)? The information-based mechanism

This study empirically examines whether and how analyst coverage drives CSR under different governance characteristics and information characteristics of firms. The results first show that analyst coverage positively relates to firms’ CSR. In addition, the results find that CEO duality, Institutional ownership, executive ownership, real earnings management, and accounting conservatism moderate the relationship between analyst coverage and CSR. This study did a lot of testing and it is easy to understand. However, I have some concerns about the contribution, hypothesis development, and writing used in the study. I elaborate on my comments below.

Major Concerns:

1. Adhikari (2016) find that firms with greater analyst coverage tend to be less socially responsible, but this paper finds that the relationship between analyst coverage and CSR was positive. This research needs to explain the reasons for the difference.

2. The author uses " The information-based mechanism " as the subtitle of the article. However, in the hypotheses part, the author did not test how can analysts reduce information asymmetry and thus affect CSR. Instead, the author states that poor corporate governance and low financial information quality increase the space for analysts to influence CSR. This description proves that corporate governance can complement the effect of analyst supervision (from the perspective of supervision rather than from the perspective of information asymmetry).

The above problems lead to two extended problems:

(1) In the introduction part, the author skipped the description of CEO duality, Institutional ownership, and executive ownership. Instead, it describes real earnings management and accounting conservatism directly after describing the impact of analysts on CSR (see page 5: The information heterogeneity, such as various accounting information characters, such as real earnings management and accounting conservatism…).

(2) H3 does not clarify why institutional ownership positively moderates the relationship between analyst coverage and CSR.

I suggest that the author clarify what governance characteristics and information characteristics actually represent and write them out clearly in hypotheses development. Do not let the reader feel that it is supervision, rather than information. Or the author can directly focus on supervision.

Minor concerns:

1. The statistics of the real earnings management and accounting conservatism variables should be shown in Tables 1 and 2.

2. The authors need to explain why this research does not use the accrual earnings as Information characteristics, or supplement the result.

3. The article has spelling or punctuation errors that need to be corrected carefully.

6. PLOS authors have the option to publish the peer review history of their article (what does this mean?). If published, this will include your full peer review and any attached files.

Reviewer #1: No

Reviewer #2: No

---

## [Author Response · Author response to Decision Letter 0]

2 Dec 2023

Response Letter

It is with excitement that we resubmit to you a revised version of manuscript “How does analyst coverage influence corporate social responsibility (CSR)? The governance- and information-based mechanisms (PONE-D-23-17772)” for the Plos One. Many thanks to all the editors for their valuable time, constructive comments, and valuable suggestions. Based on the review comments, we completed the revision of the manuscript with the full cooperation of the author team and resubmitted the new version before the deadline. We thank you for your time and details provided. We have incorporated the suggested revisions into this refined manuscript to the best of our ability. We have benefited greatly from these insightful suggested revisions. We look forward to working with you to bring the manuscript closer to publication in Plos One.

Based on the editor and reviewers’ comments and suggestions, we have carefully revised the manuscript. The following is a detailed response to the editor and reviewers’ comments.

Reviewer #1: This paper documents the positive association between analyst coverage and CSR. Authors demonstrate that both governance channel and information channel play important role in the positive association. The main contribution of this article is to document the importance of information channel.

I enjoyed reading this paper. CSR is a trendy topic, but its economic mechanism is not well studied. I also appreciate various tests authors did to support their hypotheses. However, I have some concerns on economic mechanism and the execution of the tests.

Response: Thank you very much for recognizing the revisions we've made to this before, and your comment is invaluable.

Major concerns:

1) The authors could expand on the theoretical mechanisms linking analyst coverage to CSR. In particular, how does information asymmetry moderate the relationship between coverage and CSR engagement? As authors explained, the key economic mechanism, information function, refers to ‘reliable degree of external financial information’; if firms are covered by more analyst teams, investors will pay more attention to firms’ CSR level. Does it necessarily lead to monotonical increase/decrease in CSR? Some specific examples of how reduced information asymmetry impacts CSR spending would help readers better understand the posited mechanisms. Currently, the discussion focuses more on the link between asymmetry and stock prices. Clarifying the CSR mechanism specifically would make an important contribution.

Response: Thanks a lot for your warmhearted instruction. Following your comments, we substantively revise the manuscript. For example:

In “Theoretical background and hypotheses” section, “First, analyst coverage could promote CSR by relieving agency problem and averting managers’ opportunistic behaviors. For routine tasks, analysts constantly monitor managers by analyzing, processing, and transmitting message about firm performance and probing into business strategies [24]. Despite uncertainty around CSR, external concerns from analyst could superintend the managers’ behavior and propel them to spend time, effort, and allocate resources to fulfill their duty to construct a better social community beyond the official legal obligations. The reduction in CSR investment, once observed by analysts, could result in considerable negative outcome of firm market value. To minimize the possibility of getting adverse feedbacks, managers would be less aggressive in reducing CSR investment and make earnest endeavors to achieve better CSR performance toward which analysts have favorable attitudes.

Second, the attention from analysts could ease the information asymmetry problem between firms and related parties. Due to the multidimensional nature of CSR, it is complex for general investors to estimate its real and intrinsic value. The valuation for firms will be discounted with incompletely conveyed CSR information [25]. Analysts could recognize the potential of CSR and factor it into investment recommendations [6]. Stock recommendations from financial analysts which represent a supposedly unbiased third party are valued by investors, especially in the developing country-level external governance environment [26]. While analysts not only pay attention to accounting fundamentals but also to other relevant professional and private information in poor information environments, analyst coverage is more influential in boosting CSR by exhibiting accurate assessment of social investment [15]. CSR is the key component of organizational sustainability, and CSR-related policies have become critical non-financial information for analysts when they prepare earnings forecasts. More analyst coverage could reduce information asymmetry and help to reasonably value corporate CSR. Considering the significant impact of analysts on the valuation of investors [26], managers will be pushed to take account of CSR when the related firms are followed and covered by analysts.”

In Discussion section, we highlight:

“This research enriches the prior literature by disclosing the different roles of financial analyst on CSR, by the governance channel and the information channel. On the one side, analyst coverage and firms’ inside governance mechanism complements with each other in the field of CSR. The impact of financial analyst on CSR is weaker in an inferior inside governance condition. On the other side, analyst coverage substitutes with information condition in the aspect of impacting CSR. The positive role of analyst coverage on CSR is more obvious in the worse information condition.”

“For example, the separation of CEO and chair and the participation of institutional investors are encouraged, and stock-based compensation to managers or directors needs care. The worse information situations demand analyst coverage to shape firms socially benefit behaviors more. For example, it shows that analyst coverage makes a deeper effect on CSR, in the worse information condition, specifically higher real earnings management and less accounting conservatism condition. All in all, financial analysts play different role with governance and information context. Analyst coverage complements with governance system, while substitutes for information channels to make a difference in corporate social friend activities.”

2) The monitoring and information channels seem potentially overlapping - are they mutually exclusive? Given that gauging the information channel is a key contribution, it would be helpful for the authors to further distinguish it from the monitoring channel and quantify each one's relative contribution to the empirical results. For instance, the moderating effects of earnings news and accounting conservatism are intended to provide evidence of the information channel, but may also reflect monitoring. Discussing the degree to which the findings uniquely support the information channel would further strengthen the paper's contribution.

Response: Thanks a lot for your warmhearted instruction. Following your comments, we revised the title: How does analyst coverage influence corporate social responsibility (CSR)? The governance- and information-based mechanisms

Then, we revised the Introduction section:

“In fact, investing in CSR activities is dependent on motivations of insiders such as the CEO and the board. Their evaluations on the information from analysts towards CSR are often affected by their power or discretionary decision-making. Thus, CEO duality, institutional ownership, or executive ownership could exert influence on the driving forces of analysts on corporate CSR. The concentration of the power makes CEO has the negotiating advantage among the parties and be less likely to cater to the expectations of stakeholders. Compared to the retail investors, institutional investors not only pay greater attention to various stakeholders (e.g. analysts), but also express serious concerns about societal expectations (e.g. CSR) in the oversight of managerial decision-making processes. Like CEO duality, concentrated executive ownership enlarges the power and influence of manager and could lead to the similar outcome of dual CEOs.”

In Discussion section, we highlight:

“This research enriches the prior literature by disclosing the different roles of financial analyst on CSR, by the governance channel and the information channel. On the one side, analyst coverage and firms’ inside governance mechanism complements with each other in the field of CSR. The impact of financial analyst on CSR is weaker in an inferior inside governance condition. On the other side, analyst coverage substitutes with information condition in the aspect of impacting CSR. The positive role of analyst coverage on CSR is more obvious in the worse information condition.”

“For example, the separation of CEO and chair and the participation of institutional investors are encouraged, and stock-based compensation to managers or directors needs care. The worse information situations demand analyst coverage to shape firms socially benefit behaviors more. For example, it shows that analyst coverage makes a deeper effect on CSR, in the worse information condition, specifically higher real earnings management and less accounting conservatism condition. All in all, financial analysts play different role with governance and information context. Analyst coverage complements with governance system, while substitutes for information channels to make a difference in corporate social friend activities.”

3) As authors mentioned, the result could be driven by reverse causality. Authors employ 2SLS to address endogeneity, however, there are several additional tests that could be done before authors claim causality. For example, i) industry*year fixed effect in case there is any shift in the distribution of CSR ii) matching tests given endogeneity of analyst coverage (like PSM).

Response: Thanks a lot for your warmhearted instruction. We adjust this part, as follow. First we added Regression Models. Then, for a specific firm, its behavior can hardly be the casual factor of its external institutions. While brokerage institution size is an external institutional factor and unlikely impossible linkage to CSR of the firms that the analysts incline to follow and coverage, hence, the change of analyst coverage driven by the difference of brokerage institutions could strength the causality, because it helps to mitigate the endogeneity concern. Moreover, prior literature in the vein of examining the relationship between financial analyst and innovation (He and Tian, 2013) choose the same instrumental variable. The identical measurement of the same variable, specifically analyst coverage, makes our paper be more consistent with the system of the whole literature. “Referring to the former results, an interesting phenomenon is observed that the magnitudes of the estimated coefficients of the 2SLS are fairly larger than the OLS. Accordingly, we can get that the coefficients of regressions are downwardly biased resulting by the problem of endogeneity. This conclusion implies that certain factors, which can both induce the firms less socially responsible and reduce the attention of analysts, are ignored. By mitigating the endogeneity problem of omitting variable, the effect of analysts on corporate socially benefit behaviors gets more positive. Therefore, we can get that the causal relationship between analysts and CSR are probably exist, by applying the 2SLS regression.”

Due to the time limitation, matching tests given endogeneity of analyst coverage (like PSM) will be conducted in the future. 

Minor issues:

Cross reference: Su (2019) is not cited in reference.

Response: Thanks a lot for your warmhearted instruction. We added the Su (2019) in reference.

Su, K. (2019). Does religion benefit corporate social responsibility (CSR)? Evidence from China. Corporate Social Responsibility and Environmental Management, 26(6), 1206-1221.

We also new citations of up-to-date and high-quality papers on the subjects covered in manuscript. Meanwhile, we carefully format the manuscript following the guidance of the journal, such as the headers, references, and so on. 

Thanks again for your precious time, constructive comments, and valuable suggestions. We sincerely appreciate it.

Reviewer #2: How does analyst coverage influence corporate social responsibility (CSR)? The information-based mechanism

This study empirically examines whether and how analyst coverage drives CSR under different governance characteristics and information characteristics of firms. The results first show that analyst coverage positively relates to firms’ CSR. In addition, the results find that CEO duality, Institutional ownership, executive ownership, real earnings management, and accounting conservatism moderate the relationship between analyst coverage and CSR. This study did a lot of testing and it is easy to understand. 

Response: Thank you very much for recognizing the revisions we've made to this before, and your comment is invaluable.

However, I have some concerns about the contribution, hypothesis development, and writing used in the study. I elaborate on my comments below.

Major Concerns:

1. Adhikari (2016) find that firms with greater analyst coverage tend to be less socially responsible, but this paper finds that the relationship between analyst coverage and CSR was positive. This research needs to explain the reasons for the difference.

Response: Thanks a lot for your warmhearted instruction. Following your comments, we substantively revise the theoretical logic.

In “Theoretical background and hypotheses” section, “First, analyst coverage could promote CSR by relieving agency problem and averting managers’ opportunistic behaviors. For routine tasks, analysts constantly monitor managers by analyzing, processing, and transmitting message about firm performance and probing into business strategies [24]. Despite uncertainty around CSR, external concerns from analyst could superintend the managers’ behavior and propel them to spend time, effort, and allocate resources to fulfill their duty to construct a better social community beyond the official legal obligations. The reduction in CSR investment, once observed by analysts, could result in considerable negative outcome of firm market value. To minimize the possibility of getting adverse feedbacks, managers would be less aggressive in reducing CSR investment and make earnest endeavors to achieve better CSR performance toward which analysts have favorable attitudes.

Second, the attention from analysts could ease the information asymmetry problem between firms and related parties. Due to the multidimensional nature of CSR, it is complex for general investors to estimate its real and intrinsic value. The valuation for firms will be discounted with incompletely conveyed CSR information [25]. Analysts could recognize the potential of CSR and factor it into investment recommendations [6]. Stock recommendations from financial analysts which represent a supposedly unbiased third party are valued by investors, especially in the developing country-level external governance environment [26]. While analysts not only pay attention to accounting fundamentals but also to other relevant professional and private information in poor information environments, analyst coverage is more influential in boosting CSR by exhibiting accurate assessment of social investment [15]. CSR is the key component of organizational sustainability, and CSR-related policies have become critical non-financial information for analysts when they prepare earnings forecasts. More analyst coverage could reduce information asymmetry and help to reasonably value corporate CSR. Considering the significant impact of analysts on the valuation of investors [26], managers will be pushed to take account of CSR when the related firms are followed and covered by analysts.”

2. The author uses " The information-based mechanism " as the subtitle of the article. However, in the hypotheses part, the author did not test how can analysts reduce information asymmetry and thus affect CSR. Instead, the author states that poor corporate governance and low financial information quality incre

---

## [Decision Letter · Decision Letter 1]

19 Jan 2024

PONE-D-23-17772R1How does analyst coverage influence corporate social responsibility (CSR)? The governance- and information-based mechanismPLOS ONE

Dear Dr. Du,

Thank you for submitting your manuscript to PLOS ONE. After careful consideration, we feel that it has merit but does not fully meet PLOS ONE’s publication criteria as it currently stands. Therefore, we invite you to submit a revised version of the manuscript that addresses the points raised during the review process.

**ACADEMIC EDITOR:** have completed my evaluation of your manuscript. The reviewers recommend reconsideration of your manuscript following minor revision and modification. I invite you to resubmit your manuscript after addressing their comments.

We look forward to receiving your revised manuscript.

Kind regards,

Yuan-Teng Hsu, Ph.D.

Academic Editor

PLOS ONE

Journal Requirements:

Additional Editor Comments:

I have completed my evaluation of your manuscript. The reviewers recommend reconsideration of your manuscript following minor revision and modification. I invite you to resubmit your manuscript after addressing their comments.

Reviewers' comments:

Reviewer's Responses to Questions

**Comments to the Author**

1. If the authors have adequately addressed your comments raised in a previous round of review and you feel that this manuscript is now acceptable for publication, you may indicate that here to bypass the “Comments to the Author” section, enter your conflict of interest statement in the “Confidential to Editor” section, and submit your "Accept" recommendation.

Reviewer #1: (No Response)

Reviewer #2: All comments have been addressed

2. Is the manuscript technically sound, and do the data support the conclusions?

Reviewer #1: Yes

Reviewer #2: Yes

3. Has the statistical analysis been performed appropriately and rigorously? 

Reviewer #1: Yes

Reviewer #2: Yes

4. Have the authors made all data underlying the findings in their manuscript fully available?

Reviewer #1: Yes

Reviewer #2: Yes

5. Is the manuscript presented in an intelligible fashion and written in standard English?

Reviewer #1: Yes

Reviewer #2: Yes

6. Review Comments to the Author

Reviewer #1: Thank you for revising the manuscript based on previous review. I carefully examine the revised version. While some improvements have been made, there are still some issues need to be addressed before the manuscript can be considered for publication.

About comment 1:

I appreciate authors’ effort in providing examples of positive association between analyst coverage and CSR. However, authors need to further explain the underlying mechanism of the positive association. Simply saying CSR is an important information source does not explain why higher analyst scrutiny incentivize firms to increase CSR engagement. For example, would firms significantly increase CSR investment just to send a noisy signal to the market, even if they think CSR is informative? The authors need to develop a more cogent argument for the positive sign.

About comment 2:

I recognize authors’ adjustment on the title and in hypothesis section. I am satisfied with the adjustment.

About comment 3:

I appreciate authors’ effort on constructing instrument variables like brokerage institution. I am satisfied with the revised robustness section.

In summary, while progress has been made, further revisions are still needed to address my previous concerns. I hope these comments are helpful. Please let me know if you have any further concerns.

Reviewer #2: How does analyst coverage influence corporate social responsibility (CSR)? The governance- and information-based mechanism

According to the comments of the reviewers, the authors have greatly revised the article, and it can be seen that the quality of the article has been significantly improved. However, I think there are still some contents in the article that need to be modified. Here are my comments:

1. In the title, using the word "mechanism" to represent "moderators" is considered unprofessional. Few top journals use this expression, and the authors should avoid the use of "Chinglish."

2. The authors mentioned in introduction that "few studies have disclosed the impact of external analysts' attention on a firm's CSR and its underlying information mechanisms ", but I think this description is inappropriate, there is already a lot of relevant literature.

3.In introduction, the authors list some of the literature, then mentioned that” However, as the theorical analysis implied in the prior literature, analyst coverage may work on CSR through the governance- and information-based mechanism”, however, it is not obvious to me that this conclusion can be drawn from the preceding literature.

From the sentence " In fact, investing in CSR activities is dependent on motivations of insiders such as the CEO and the board. Their evaluations on the information from analysts towards CSR are often affected by their power or discretionary decision-making.", I also cannot conclude why CEO duality, institutional ownership, or executive ownership could exert influence on the driving forces of analysts on corporate CSR. The authors need to pay attention to the logic when using connectionists such as "in fact" and "thus".

4. Before the sentence " As a result, the comparisons of the moderate effect of corporate governance and external financial information are necessary. " The author should explain the reasons for using financial information as moderator in more detail.

5.In Hypothesis 3, the authors do not well describe why institutional ownership affect the relationship between analysts and CSR, but spend a lot of words on how institutional investors affect CSR, which does not have much relevance to the hypothesis. At the same time, the author mentioned that " Therefore, the firms have more motivation to engage in CSR when face high institutional ownership and financial coverage. " would make readers question the causal relationship between analysts and CSR, which is not in line with the title of this paper " How does analyst coverage influence corporate social responsibility ". Similarly, in Hypothesis 4-6, authors need to add the description of why executive ownership/earnings management/conservatism affects the relationship between analysts and CSR (Instead of just describing why executive ownership/earnings management/conservatism affects CSR).

5.In Data and samples, the authors need to describe why only manufacturing firms was selected.

6. The authors need to explain why this research does not use the accrual earnings as Information characteristics, or supplement the result.

7. PLOS authors have the option to publish the peer review history of their article (what does this mean?). If published, this will include your full peer review and any attached files.

Reviewer #1: No

Reviewer #2: No

---

## [Author Response · Author response to Decision Letter 1]

3 Mar 2024

Response Letter

Dear Editor and Reviewers,

It is with excitement that we resubmit to you a revised version of manuscript “How does analyst coverage influence corporate social responsibility (CSR)? The governance- and information-based perspectives (PONE-D-23-17772)” for the Plos One. Many thanks to all the editors for their valuable time, constructive comments, and valuable suggestions. Based on the review comments, we completed the revision of the manuscript with the full cooperation of the author team and resubmitted the new version before the deadline. We thank you for your time and details provided. We have incorporated the suggested revisions into this refined manuscript to the best of our ability. We have benefited greatly from these insightful suggested revisions. We look forward to working with you to bring the manuscript closer to publication in Plos One.

Based on the editor and reviewers’ comments and suggestions, we have carefully revised the manuscript. The following is a detailed response to the editor and reviewers’ comments.

Reviewer #1: Thank you for revising the manuscript based on previous review. I carefully examine the revised version. While some improvements have been made, there are still some issues need to be addressed before the manuscript can be considered for publication.

About comment 1:

I appreciate authors’ effort in providing examples of positive association between analyst coverage and CSR. However, authors need to further explain the underlying mechanism of the positive association. Simply saying CSR is an important information source does not explain why higher analyst scrutiny incentivize firms to increase CSR engagement. For example, would firms significantly increase CSR investment just to send a noisy signal to the market, even if they think CSR is informative? The authors need to develop a more cogent argument for the positive sign.

Response: Thanks a lot for your warmhearted instruction. Following your comments, we revised the argument for the hypotheses (see the revised manuscript with track changes) and hope to meet your high expectations.

About comment 2:

I recognize authors’ adjustment on the title and in hypothesis section. I am satisfied with the adjustment.

Response: Thank you very much for recognizing the revisions we've made to this before, and your comment is invaluable.

About comment 3:

I appreciate authors’ effort on constructing instrument variables like brokerage institution. I am satisfied with the revised robustness section.

Response: Thank you very much for recognizing the revisions we've made to this before, and your comment is invaluable.

In summary, while progress has been made, further revisions are still needed to address my previous concerns. I hope these comments are helpful. Please let me know if you have any further concerns.

Thanks again for your precious time, constructive comments, and valuable suggestions. We sincerely appreciate it.

Reviewer #2: How does analyst coverage influence corporate social responsibility (CSR)? The governance- and information-based mechanism

According to the comments of the reviewers, the authors have greatly revised the article, and it can be seen that the quality of the article has been significantly improved. However, I think there are still some contents in the article that need to be modified. Here are my comments:

1. In the title, using the word "mechanism" to represent "moderators" is considered unprofessional. Few top journals use this expression, and the authors should avoid the use of "Chinglish."

Response: Thanks a lot for your warmhearted instruction. Following your comments, we revised the title: “How does analyst coverage influence corporate social responsibility (CSR)? The governance- and information-based perspectives”.

2. The authors mentioned in introduction that “few studies have disclosed the impact of external analysts' attention on a firm's CSR and its underlying information mechanisms”, but I think this description is inappropriate, there is already a lot of relevant literature.

Response: Thanks a lot for your warmhearted instruction. Following your comments, we revised the expression: “Recently, a strand of attention has been paid to disclosed the impact of external analysts’ attention on a firm’s CSR and its underlying mechanisms [13].”

3.In introduction, the authors list some of the literature, then mentioned that “However, as the theorical analysis implied in the prior literature, analyst coverage may work on CSR through the governance- and information-based mechanism”, however, it is not obvious to me that this conclusion can be drawn from the preceding literature. From the sentence “In fact, investing in CSR activities is dependent on motivations of insiders such as the CEO and the board. Their evaluations on the information from analysts towards CSR are often affected by their power or discretionary decision-making.", I also cannot conclude why CEO duality, institutional ownership, or executive ownership could exert influence on the driving forces of analysts on corporate CSR. The authors need to pay attention to the logic when using connectionists such as "in fact" and "thus".

Response: Thanks a lot for your warmhearted instruction. Following your comments, we revised the expression and adjusted the logic: “These studies offer evidence that financial analysts could exert influence on CSR through their governance or information related functions. In practical, investing in CSR activities is dependent on motivations of insiders such as the CEO and the board. Their evaluations on the information from analysts towards CSR are often affected by their power or discretionary decision-making. The concentration of the power makes CEO has the negotiating advantage among the parties and be less likely to cater to the expectations of stakeholders. Compared to the retail investors, institutional investors not only pay greater attention to various stakeholders (e.g. analysts), but also express serious concerns about societal expectations (e.g. CSR) in the oversight of managerial decision-making processes. Like CEO duality, concentrated executive ownership enlarges the power and influence of manager and could lead to the similar outcome of dual CEOs. Thus, CEO duality, institutional ownership, or executive ownership might be the driving forces of analysts on corporate CSR to a certain extent.”

4. Before the sentence “As a result, the comparisons of the moderate effect of corporate governance and external financial information are necessary.” The author should explain the reasons for using financial information as moderator in more detail.

Response: Thanks a lot for your warmhearted instruction. Following your comments, we added explanations for supplementing the moderators:

“Additionally, the information heterogeneity, such as various accounting information characters (e.g. real earnings management and accounting conservatism), may result in a different impact of analyst coverage on CSR, which is scarcely empirically examined in the prior studies. Meanwhile, both real earnings management and accounting conservatism reflect the reliable degree of external financial information [19, 20]. Real earnings management emphasizes whether the accounting information is manipulated through real activities, such as adjusting production and producing cost. Accounting conservatism positively relates to the timeliness of confirming bad news, while negatively collated to good news. The process of considering financial analysts’ opinions into CSR behaviors cannot ignore the necessary conditions which comprise reactions from the capital market. It is necessary to specially disclose the role of external financial information in corporate strategies towards CSR investments. While the characteristics of external financial information is significant different from corporate governance system, the comparisons of the effects of corporate governance and external financial information are also relevant.”

5.In Hypothesis 3, the authors do not well describe why institutional ownership affect the relationship between analysts and CSR, but spend a lot of words on how institutional investors affect CSR, which does not have much relevance to the hypothesis. At the same time, the author mentioned that “Therefore, the firms have more motivation to engage in CSR when face high institutional ownership and financial coverage.” would make readers question the causal relationship between analysts and CSR, which is not in line with the title of this paper” How does analyst coverage influence corporate social responsibility”. Similarly, in Hypothesis 4-6, authors need to add the description of why executive ownership/earnings management/conservatism affects the relationship between analysts and CSR (Instead of just describing why executive ownership/earnings management/conservatism affects CSR).

Response: Thanks a lot for your warmhearted instruction. Following your comments, we revised the argument for the hypotheses. For example: 

“Being a strong symbol of prosocial responsibility, CSR naturally is regarded as an important indicator for investment decisions. Institutional investors often have a shared understanding of the value of CSR with analysts for the following firms.” in Hypothesis 3.

“Firms with high executive ownership often have less motivations to engage in CSR although analysts are encouraging corporate investments in CSR. For example, firms cut off charitable donation after an increase in executive ownership [37]. Thus, a high level of executive ownership will lead to the constraint of analysts and pay less attention to analysts’ opinions, such as prioritizing CSR.” in Hypothesis 4.

“In other words, financial analysts play a more profound role on CSR in firms with larger degree of real earnings management. Firms naturally have more motivations to engage in CSR to go along with analysts when they have special preference to better CSR performance.” in Hypothesis 5.

“In other words, accounting conservatism may be substitute with financial analysts by regulating the agent parties. The value guidance from analysts is likely to be given a smaller weight in this event. Although analysts have special preference to CSR, managers often choose to turn a blind eye to these opinions.” in Hypothesis 6.

5. In Data and samples, the authors need to describe why only manufacturing firms was selected.

Response: Thanks a lot for your comments. Manufacturing firms have been chosen because they are the mainspring of economy development and the main sources of air, land, and water pollution in China. The research on their CSR is quite relevant for the sustainability of the country. 

6. The authors need to explain why this research does not use the accrual earnings as Information characteristics, or supplement the result.

Response: Thanks a lot for your comments. In this study, we did not use accrual earnings as information characteristics because we think real earnings management is more difficult to detect than accrual-based earnings management and is constrained by analyst following (Enomoto et al., 2015). The influence of analyst coverage on CSR would be more sensitive by employing real earnings management as the information characteristics. 

Enomoto, M., Kimura, F., & Yamaguchi, T. (2015). Accrual-based and real earnings management: An international comparison for investor protection. Journal of Contemporary Accounting & Economics, 11(3), 183-198.

Thanks again for your precious time, constructive comments, and valuable suggestions. We sincerely appreciate it.

---

## [Decision Letter · Decision Letter 2]

28 Mar 2024

How does analyst coverage influence corporate social responsibility (CSR)? The governance- and information-based perspectives

PONE-D-23-17772R2

Dear Dr. Du,

We’re pleased to inform you that your manuscript has been judged scientifically suitable for publication and will be formally accepted for publication once it meets all outstanding technical requirements.

Kind regards,

Yuan-Teng Hsu, Ph.D.

Academic Editor

PLOS ONE

Additional Editor Comments (optional):

I have confirmed the current modified version and the response to the reviewer. I think it is suitable for publication.

Reviewers' comments:

Reviewer's Responses to Questions

**Comments to the Author**

1. If the authors have adequately addressed your comments raised in a previous round of review and you feel that this manuscript is now acceptable for publication, you may indicate that here to bypass the “Comments to the Author” section, enter your conflict of interest statement in the “Confidential to Editor” section, and submit your "Accept" recommendation.

Reviewer #1: All comments have been addressed

Reviewer #2: (No Response)

2. Is the manuscript technically sound, and do the data support the conclusions?

Reviewer #1: Yes

Reviewer #2: (No Response)

3. Has the statistical analysis been performed appropriately and rigorously? 

Reviewer #1: Yes

Reviewer #2: (No Response)

4. Have the authors made all data underlying the findings in their manuscript fully available?

Reviewer #1: No

Reviewer #2: (No Response)

5. Is the manuscript presented in an intelligible fashion and written in standard English?

Reviewer #1: Yes

Reviewer #2: (No Response)

6. Review Comments to the Author

Reviewer #1: All comments has been addressed and I am satisfied with this version of manuscript. I appreciate all the changes been made and wish you good luck in publication!

Reviewer #2: (No Response)

7. PLOS authors have the option to publish the peer review history of their article (what does this mean?). If published, this will include your full peer review and any attached files.

Reviewer #1: No

Reviewer #2: No
